# Comparative transcriptomic analysis reveals translationally relevant processes in mouse models of malaria

**Athina Georgiadou[1,2†], Claire Dunican[1,2†], Pablo Soro-Barrio[1‡], Hyun Jae Lee[3§], Myrsini Kaforou[1,2], Aubrey J Cunnington[1,2]***

[1]Section of Paediatric Infectious Disease, Department of Infectious Disease, Imperial College London, London, United Kingdom; [2]Centre for Paediatrics and Child Health, Imperial College London, London, United Kingdom; [3]Institute for Molecular Bioscience, University of Queensland, Brisbane, Australia

**\*For correspondence:**
a.cunnington@imperial.ac.uk

[†]These authors contributed equally to this work

**Present address:** [‡]The Francis Crick Institute, London, United Kingdom; [§]The Peter Doherty Institute for Infection and Immunity, Melbourne, Australia

**Competing interest:** The authors declare that no competing interests exist.

**Abstract** Recent initiatives to improve translation of findings from animal models to human disease have focussed on reproducibility but quantifying the relevance of animal models remains a challenge. Here, we use comparative transcriptomics of blood to evaluate the systemic host response and its concordance between humans with different clinical manifestations of malaria and five commonly used mouse models. *Plasmodium yoelii* 17XL infection of mice most closely reproduces the profile of gene expression changes seen in the major human severe malaria syndromes, accompanied by high parasite biomass, severe anemia, hyperlactatemia, and cerebral microvascular pathology. However, there is also considerable discordance of changes in gene expression between the different host species and across all models, indicating that the relevance of biological mechanisms of interest in each model should be assessed before conducting experiments. These data will aid the selection of appropriate models for translational malaria research, and the approach is generalizable to other disease models.

## Editor's evaluation

Using comparative transcriptomics, the authors performed an unbiased investigation of similarities and differences of mouse malaria models and human *Plasmodium falciparum* malaria. Whilst the data cannot convincingly identify which mouse models are best suited for studying specific human malaria phenotypes, the comparative analyses do indicate that these models reflect the broad diversity of human malaria disease. These comparative analyses provide a scientific rationale for the use of rodent malaria models.

## Introduction

Animal models have played an important role in current understanding and treatment of many human diseases. Historically animal models were often selected because they reproduced certain clinical or pathological features of human disease (**Hau, 2008**), and their use has often been reinforced when treatments effective in the model were found to be effective in humans. However, this approach has limitations, because the same clinical or pathological features can occur as a result of different biological processes, and mechanisms that may be important in human disease might not be recapitulated or may be redundant in animal models, and vice-versa (**Pound and Ritskes-Hoitinga, 2018**; **Justice and Dhillon, 2016**). A fundamental and largely unresolved question is how best to define or quantify

the relevance of any given animal model to the corresponding human disease (*Justice and Dhillon, 2016*; *Ferreira et al., 2019*).

Mice are the most widely used model animals for many diseases, including infectious diseases, and for study of corresponding protective or pathogenic immune responses. Mouse models have significantly broadened our understanding of the function and structure of mammalian immune systems and disease mechanisms. Despite the evolutionary distance between human and mouse (*Mestas and Hughes, 2004*) and the high evolutionary pressure on immune systems (*Cagliani and Sironi, 2013*), the principles of the immune systems for these species remain remarkably similar. However, there are also numerous differences between mice and humans in terms of their response to infection (*Mestas and Hughes, 2004*). Therefore, it is inevitable that mouse models of infection will not recapitulate all features of the human response, and this should be taken into account when using models to make inferences about mechanisms of human disease. Recently, we and others proposed that unbiased approaches to assessment of the host response to infection, such as comparison of transcriptomic responses, might provide a meaningful way to quantify similarities between mouse models and human disease, to assess the relevance of the models, and to aid the selection of the best models for specific hypothesis testing (*Lee et al., 2018a*).

The relevance of mouse models for translational research on the pathogenesis of severe malaria (SM) has been particularly controversial and has polarized the malaria research community (*Craig et al., 2012*). There are many different mouse malaria models, with very different characteristics dependening on the combination of parasite species (and strain) and mouse strain which are used (*Lamb et al., 2006*). Superficially these models can, between them, reproduce almost all the clinical manifestations of human SM, such as coma, seizures, respiratory distress, and severe anemia (SA) (*Zuzarte-Luis et al., 2014*). Nevertheless, there are also notable differences to human disease, such as the lack of the pathognomonic cytoadhesive sequestration of large numbers of parasite-infected red cells in the cerebral microvasculature in mice with cerebral malaria-like illness (experimental cerebral malaria, ECM) (*Ghazanfari et al., 2018*). In C57BL/6 mice infected with *Plasmodium berghei* ANKA, ECM is dependent on recruitment of CD8+ T cells to the brain, a phenomenon that was recently shown to also occur in human cerebral malaria (*Riggle et al., 2020*). C57BL/6 mice infected with *P. berghei* NK65 develop acute lung injury with similarities to malaria-associated acute respiratory distress syndrome (MA-ARDS), associated with hemozoin accumulation (*Deroost et al., 2013*), endothelial activation, and alveolar edema (*Zuzarte-Luis et al., 2014*; *Van den Steen et al., 2010*; *Claser et al., 2019*). SA can occur in C57BL/6 mice infected with all of the most commonly used mouse malaria parasite species (*P. chabaudi* AS, *P. yoelii* 17XL, *P. yoelii* 17XNL, *P. berghei* ANKA, and *P. berghei* NK65; *Thawani et al., 2014*; *Couper et al., 2007*; *Niikura et al., 2008*), sharing features with human SM anemia such as hemolysis, hemozoin-, and inflammatory cytokine-mediated suppression of erythropoiesis (*Thawani et al., 2014*; *Lamikanra et al., 2007*).

Many host-directed treatments for SM have been effective in mice, but none have yet translated into benefit in human studies, which has been considered by some as evidence that mechanisms of disease in mouse models are of little relevance to human disease (*White et al., 2010*). We contend that this polarization of views is unhelpful, and that mouse models are likely to be useful for understanding human malaria, so long as they are used selectively with full recognition of their limitations. Such limitations include: mice are not the natural hosts for the commonly used rodent malaria parasites (*Lamb et al., 2006*) natural infection of humans occurs through sporozoite inoculation during mosquito feeding (*Rénia and Goh, 2016*), but mice are often infected by injection of blood-stage parasites to ensure a reproducible inoculum of parasites (*Craig et al., 2012*) malaria naïve mice are typically used in experiments, whereas most human malaria infections occur in endemic settings where individuals have had previous malaria infections (*Doolan et al., 2009*) mice often tolerate higher parasitemias than those seen in human infections (*Fontana et al., 2016*).

In order to provide a more quantitative framework to understand how well mouse malaria models recapitulate the biological processes occurring in human malaria, and to aid selection of the most appropriate models for study of specific mechanisms of disease, we present an unbiased investigation of the similarities and differences in the host response between human malaria and mouse models using comparative transcriptomics. We demonstrate that this approach allows us to identify mouse models with the greatest similarity of host response to specific human malaria phenotypes, and that models selected in this way do indeed have similar clinical and pathological features to those of the

corresponding human phenotype. We propose that this approach should be applied more broadly to the selection of the most relevant animal models for study of malaria and other human diseases.

## Results

### Mouse models of malaria

The five rodent malaria parasite strains used in this study produce different kinetics of parasitemia, different rates of progression of illness (*Figure 1*), and different disease manifestations. Eight-week-old C57BL/6J mice infected with *P. berghei* ANKA, *P. yoelii* 17XL, and *P. berghei* NK65 developed severe illness with ascending parasitemia, consistent with previously reported outcomes of these lethal infections (*White et al., 2010*; *Carroll et al., 2010*; *Walliker et al., 1976*; *Vandermosten et al., 2018*). Humane endpoints were reached at days 8–9 in *P. berghei* ANKA, day 5 in *P. yoelii* 17XL, and day 20 in *P. berghei* NK65. Mice infected with *P. berghei* ANKA showed typical features of ECM as assessed by Rapid Murine Coma and Behavior Scale (RMCBS) scores <12 (*Carroll et al., 2010*) and by histopathology. Mice infected with *P. yoelii* 17XL developed a rapidly progressive, severe infection with hyperparasitemia (*Walliker et al., 1976*). Mice infected with *P. berghei* NK65 developed a biphasic illness with a transient recovery of initial weight loss before progression to fatal outcome in a second phase (*Vandermosten et al., 2018*).

Eight-week-old C57BL/6J mice infected with *P. yoelii* 17XNL and *P. chabaudi* AS, which lead to self-resolving infections, developed only mild symptoms as expected (*Wijayalath et al., 2014*; *Achtman et al., 2003*). Maximum severity was reached around day 14 in *P. yoelii* 17XNL and day 13 in *P. chabaudi* AS.

### Comparative analysis of infection-associated changes in gene expression

To objectively assess how similar disease-associated systemic processes occurring in mouse malaria models are to those occurring in human *P. falciparum* malaria, we used a comparative transcriptomic approach focussed on blood. Rather than directly comparing the expression of orthologous genes in humans and mice, which would be confounded by species-specific differences in constitutive gene expression, we first identified differentially expressed genes in pairwise within-species comparisons and then used these differentially expressed genes as the basis for between-species comparisons (*Figure 2a*). This also enabled us to conduct within-species adjustment for variation in leukocyte cell mixture (*Supplementary file 1A*), which is an important confounder in whole blood gene expression analysis (*Lee et al., 2018b*). Additionally, this allows for the removal of platform-specific effects, which is especially relevant for comparisons between data generated by microarray and RNA-Seq.

Sometimes we may wish to investigate the host immune response to infection per se or alternatively we may want to investigate the processes associated with severe disease pathogenesis, and these different aims require different comparator groups. In the former situation, changes in gene expression associated with infection per se are best characterized by comparison between healthy and infected states, whereas in the latter situation it may be more appropriate to compare severe and non-severe infection states.

To investigate concordance of the host response to uncomplicated malaria (UM) in humans and mice, we first focussed on comparisons between subjects with UM and healthy uninfected subjects. To assess changes in gene expression due to naturally acquired *P. falciparum* malaria, we used two human transcriptomic data sets previously published by *Idaghdour et al., 2012* and *Boldt et al., 2019*, each of which included a healthy uninfected group and an uncomplicated *P. falciparum* malaria group (*Supplementary file 1B*). As an additional comparison with infection in malaria naïve humans, we used a previously published data set from controlled human malaria infection (CHMI) in malaria-naïve adults (*Milne et al., 2021*) before infection and on the day of first symptoms. For mice, we identified changes in gene expression occurring between healthy uninfected control mice and infected mice at first onset of visible signs of illness.

To reduce confounding by infection-induced changes in the relative proportions of different leukocyte populations and large differences in leukocyte proportions between humans and mice (*Supplementary file 1A*), all primary differential expression analyses were performed with adjustment for the proportions of the major leukocyte populations (see Materials and methods; unadjusted results

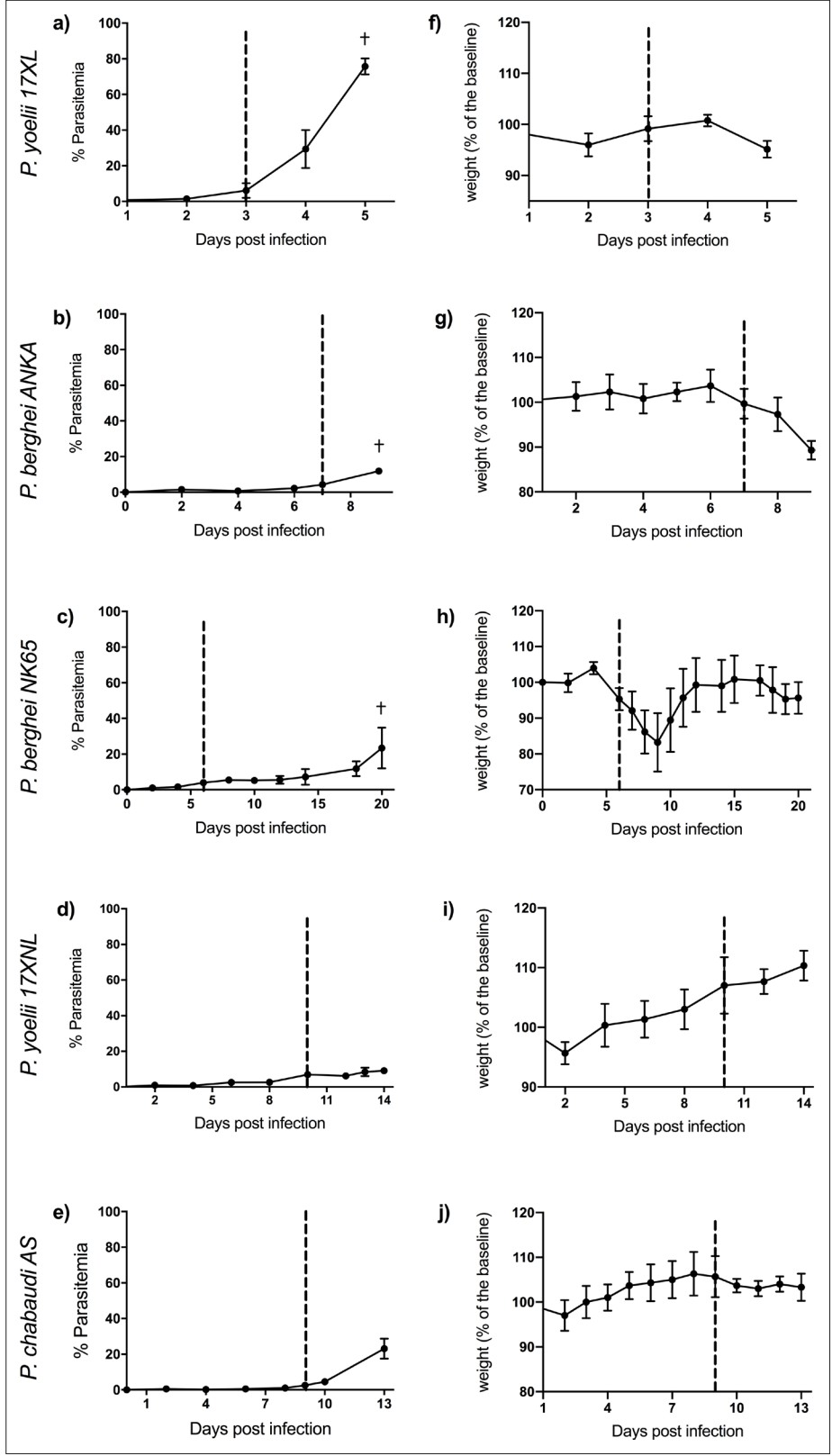

**Figure 1.** Course of infection in five mouse malaria models. Comparison of parasitemia (**a–e**) and change in weight (as percentage of baseline weight) (**f–j**) for 8-week-old C57BL/6J female wild-type mice infected with: *Plasmodium yoelii* 17XL, *P. berghei* ANKA, *P. berghei* NK65, *P. yoelii* 17XNL, and *P. chabaudi* AS, respectively. Points show mean, and bars show SD, for n=6 mice (up to and including time point of first signs of ill health, dashed vertical line) and

*Figure 1 continued on next page*

*Figure 1 continued*

n=3 mice (after dashed vertical line) for each infection. † indicates humane endpoint for lethal infections. Severity scoring for each infection shown in *Figure 1—figure supplement 1*, and individual mouse parasitemia and weights shown in *Figure 1—source data 1*.

The online version of this article includes the following source data and figure supplement(s) for figure 1:

**Source data 1.** Individual mouse parasitemia and weights.

**Figure supplement 1.** Severity scoring.

are also available for reference in *Supplementary files 2-5*). Genes with absolute log-fold change in expression >1 in the human healthy control vs UM comparison (*Supplementary files 6 and 7*) and their mouse orthologs (*Supplementary file 8*) were used for comparison between species.

First, considering only whether genes were upregulated or downregulated by infection in the mouse models, we found the mouse models varied from 58% to 73% concordance (*Supplementary file 9*, *Figure 2*) with the upregulation or downregulation in the human subjects in the study by Idaghdour et al. However, we reasoned that the relative magnitude of changes in gene expression is also important to identify the mouse models which most closely recapitulate the changes in gene expression in human malaria. To assess this, we ranked genes to account for the relative magnitude of change in expression in human malaria, and then performed a principal component analysis (PCA) with rank-weighted changes in expression (see Materials and methods). This revealed variation between the mouse models, but no model was clearly much more representative of the changes in gene expression in human UM than any other (*Figure 2b*). Indeed, when we focused only on the expression of the 20 most differentially expressed genes in human UM, we found that the mouse models showed broadly similar patterns of changes in gene expression (*Figure 2c*). When we examined the concordance of upregulation and downregulation of gene expression between the mouse models and human malaria in the Boldt et al. data set, we found less overall similarity between species in the direction of changes in gene expression (*Supplementary file 9*). Despite this, and different genes driving the axes of variation, the PCA plots revealed a remarkably similar pattern to the analysis based on the Idaghdour et al. data set, and none of the mouse models appeared to be clearly more representative of human UM than any other when accounting for the magnitude of changes in expression (*Figure 2d*). Considering the most differentially expressed genes, there was more heterogeneity in the pattern of expression (*Figure 2e*) which may be partly explained by the substantially smaller size, and analysis of pooled samples in the Boldt et al. study. Comparing differential gene expression from the mice with that of malaria naïve humans undergoing CHMI (*Supplementary file 10*) resulted in PCA plots broadly similar to those derived using the naturally acquired UM subjects (*Supplementary file 1C*), yielding similar inferences about concordance between uncomplicated or early stage infections in mice and humans.

Gene ontology (GO) analysis was used to examine the genes driving the axes of variation between humans and mouse models in the PCA plots. For the Idaghdour et al. data set, we found that PC1 showed enrichment of leukocyte mediated immunity and adaptive immune response, while PC2 showed enrichment for intrinsic apoptotic signaling in response to oxidative stress and regulation of T cell activation (*Supplementary file 11*). For the Boldt et al. data set comparison, we found that that PC1 showed enrichment of cytokine-mediated signaling pathways and hemopoiesis as the top GO terms, while for PC2 the top GO terms included immune system process and myeloid cell development (*Supplementary file 11*). These pathways are consistent with well-characterized aspects of the early host response to malaria, in which parasites are sensed by pattern recognition receptors, promoting the production of cytokines (*Gowda and Wu, 2018*), and ensuing mobilization of early myeloid progenitors from bone marrow to establish emergency myelopoiesis in the spleen (*Nahrendorf et al., 2021*; *Belyaev et al., 2013*). Proinflammatory and immunoregulatory cytokines play important roles in shaping T-cell activation and adaptive immune responses (*Urban et al., 2005*). Reactive oxygen species produced by phagocytic cells in response to parasites, and through cell-free heme released during hemolysis, contribute to inflammation and tissue damage (*Vasquez et al., 2021*).

In the Milne et al. CHMI data set, the top GO terms for PC1 were related to bacterial and interferon-γ responses, whilst PC2 was related to viral and interferon-γ responses (*Supplementary file 11*). These again are consistent with the earliest innate responses to malaria parasites, with timing and

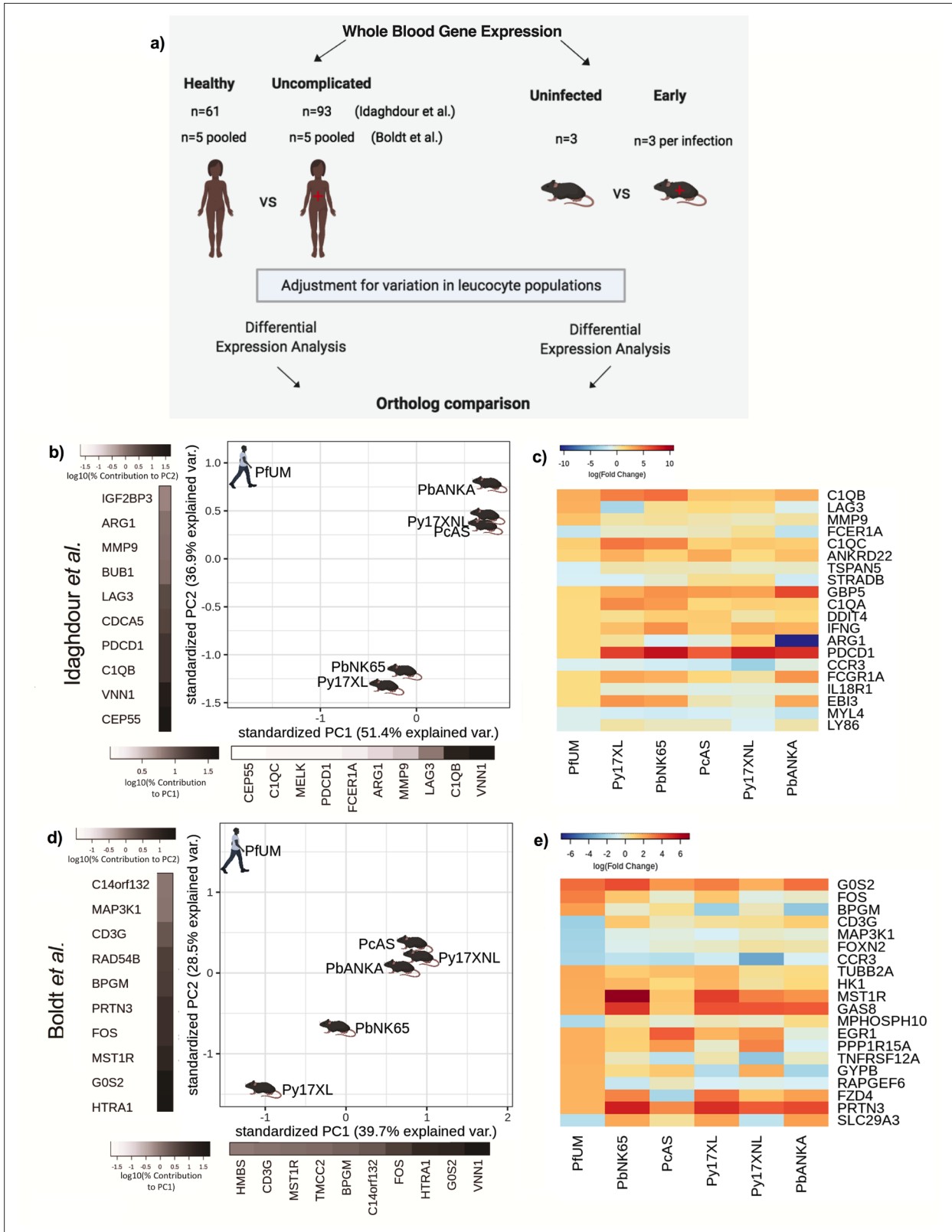

**Figure 2.** Comparison of host differential gene expression in human uncomplicated malaria and early stage illness in five mouse malaria models. (**a**) Schematic illustration of the comparative transcriptomic analysis. (**b, d**) Principal component analysis (PCA) plots generated using rank-normalized log-fold change (logFC) values from the human and mouse differential expression analyses. Only genes with 1:1 mouse and human orthologs and with absolute logFC value greater than 1 in the corresponding human comparison were included. Comparison of changes in gene expression in the mouse

*Figure 2 continued*

models (uninfected vs. early in infection, *Supplementary file 12*) with those in uncomplicated malaria versus healthy (PfUMH) Beninese children (**b**, *Idaghdour et al., 2012*) or Gabonese children (*Boldt et al., 2019*). The percentage of the total variation explained by principal components 1 and 2 are shown in the axis labels. Greyscale heatmaps parallel to each axis show the contributions of the 10 genes contributing most to the corresponding PC. (**c, e**) Heatmaps show logFC for the 20 genes with the greatest absolute logFC values in the human differential gene expression analysis, and their orthologs in each mouse model, corresponding to the analyses illustrated in (**b**) and (**d**), respectively. Mouse models are ordered left to right in order of increasing dissimilarity to the human disease, based on the Euclidian distance calculated from all principal components (*Supplementary file 13*). The rows (genes) are ordered by absolute log-fold change in the human comparison in descending order. n=3 for early and n=3 for late time point in each mouse model; n=93 UM, n=61 controls (Beninese children, Idaghdour et al.), n=5 pools UM and n=5 pools healthy control samples (each pool contained RNA from four Gabonese children with the same phenotype, Boldt et al.). Full heatmaps for the expression of genes contributing most to the first two principal components in humans and each mouse model shown in *Figure 2—figure supplements 1–4*. The mouse model abbreviations are as follows: PbNK65 (*P. berghei* NK65), PbANKA (*P. berghei* ANKA), PcAS (*P. chabaudi* AS), Py17XL (*P. yoelii* 17XL), and Py17XNL (*P. yoelii* 17XNL).

The online version of this article includes the following figure supplement(s) for figure 2:

**Figure supplement 1.** Additional heatmaps for *Figure 2*.

**Figure supplement 2.** Additional heatmaps for *Figure 2*.

**Figure supplement 3.** Additional heatmaps for *Figure 2*.

**Figure supplement 4.** Additional heatmaps for *Figure 2*.

duration of IFN-γ production being important determinants of whether its effects are protective or pathogenic (*King et al., 2015*; *Walther et al., 2009*; *Hermsen et al., 2003*; *De Souza et al., 1997*; *Mitchell et al., 2005*; *Amani et al., 2000*; *Villegas-Mendez et al., 2012*).

## Comparative analysis of severe malaria-associated changes in gene expression

A common approach to identify processes associated with the pathogenesis of severe infection is to compare individuals with severe manifestations against other individuals who have the same infection but have not developed severe illness (*Lee et al., 2018a*). This approach is expected to enrich for genes involved in the pathogenesis of severe illness from amongst the larger set of genes involved in the overall response to infection (*Lee et al., 2018a*). Therefore, we identified changes in gene expression in mice between the first time point at which mice developed signs of illness (early) and the maximum severity (late time point) of each of the five infection models. We compared these changes in gene expression in mice with those we had previously identified in Gambian children with UM and three different *P. falciparum* (SM) phenotypes (hyperlactatemia [HL], cerebral malaria [CM], or the combined phenotype of hyperlactatemia with cerebral malaria [CH]) (*Lee et al., 2018b*; *Figure 3a*). All differential expression analyses were performed with adjustment for the proportions of the major leukocyte populations in blood (see Materials and methods; *Supplementary file 14*), but unadjusted results are also available for reference (*Supplementary files 2 and 15*).

Overall, the direction of changes in gene expression in the mouse models were less concordant with those in human SM phenotypes than we observed in the comparisons with UM (*Supplementary file 9*, *Figure 3*). There was, however, much clearer variation between the different mouse models in how closely the changes in expression of individual genes recapitulated those observed in each human SM manifestation (*Figure 3b, d and f*). Using the principal component-based approach to compare weighted changes in gene expression in each infection, we were able to identify the models with greatest similarity to the transcriptional host response of each human SM phenotype (*Figure 3b, d and f* and *Supplementary file 9*). It is notable that even amongst the 20 most differentially expressed genes associated with each human SM manifestation, there was considerable variation in the degree of concordance and discordance with the mouse models (*Figure 3c, e and g*).

Hyperlactatemia is a relatively common manifestation of SM in children, and an independent predictor of death (*Krishna et al., 1994*). PCA revealed that *P. yoelii* 17XL and *P. berghei* NK65 models most closely recapitulated the changes in gene expression associated with this disease phenotype in Gambian children (*Figure 3b*). We performed GO enrichment analysis on the genes contributing most to the principal components explaining the greatest proportion of variation between the mouse models and human disease, identifying neutrophil degranulation driving PC1 and myeloid leukocyte activation driving PC2 (*Supplementary file 11*). Despite *P. yoelii* 17XL having the closest proximity to human malaria hyperlactatemia in the PCA plot, it was clear that even for this model many of the most

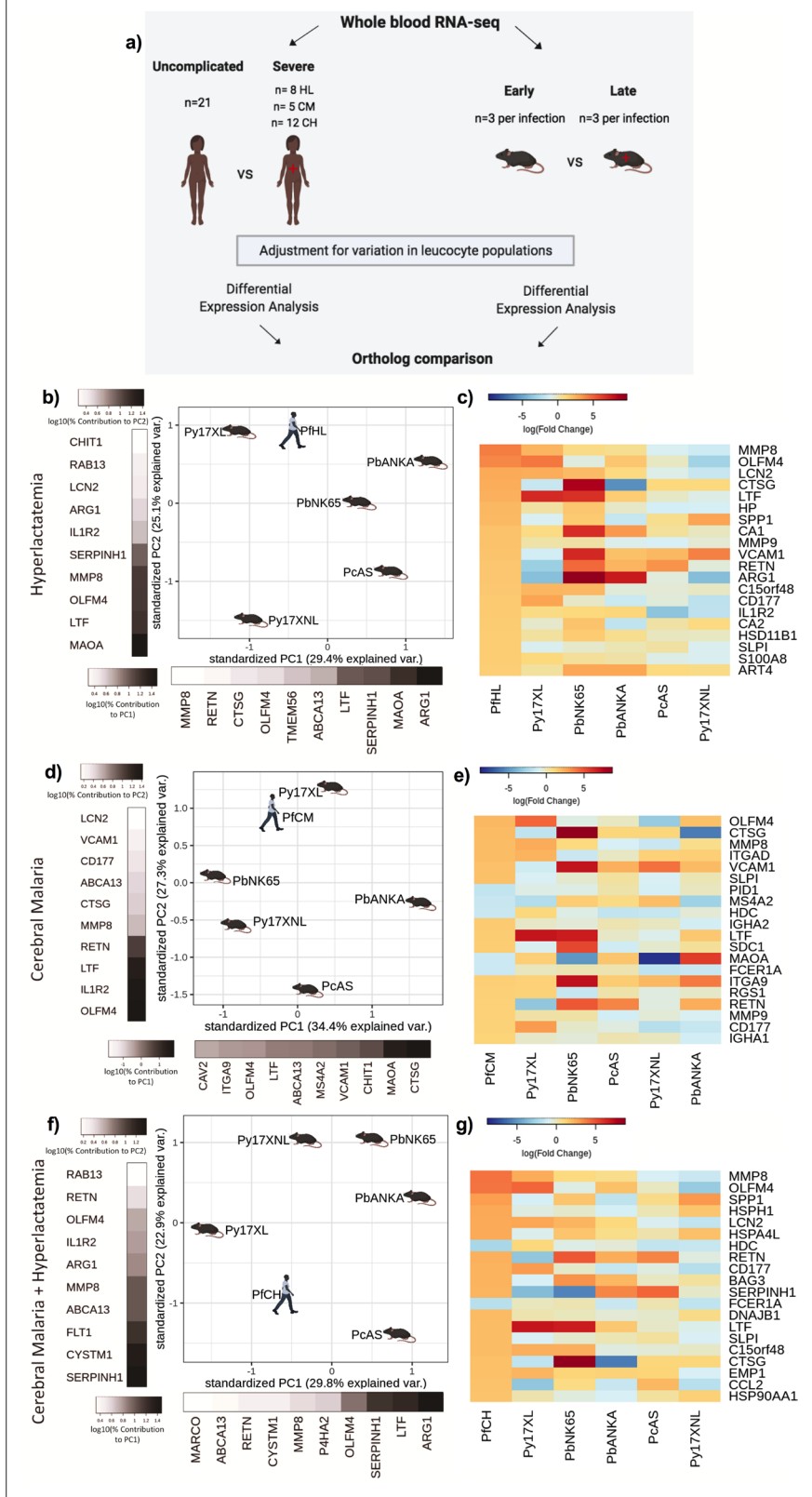

**Figure 3.** Comparison of host differential gene expression in three severe malaria phenotypes in Gambian Children and five mouse malaria models. (**a**) Schematic illustration of the comparative transcriptomic analysis. (**b, d, f**) Principal component analysis (PCA) plots generated using rank-normalized log-fold change values from the human and mouse differential expression analyses. Only genes with 1:1 mouse and human orthologs and

*Figure 3 continued on next page*

*Figure 3 continued*

with absolute logFC value greater than 1 in the corresponding human comparison were included. Comparison of changes in gene expression in the mouse models with those in human hyperlactatemia (PfHL) (**b**), cerebral malaria (PfCM) (**d**), or human hyperlactatemia plus cerebral malaria phenotype (PfCH) (**f**). The percentage of the total variation explained by principal components 1 and 2 are shown in the axis labels. Grayscale heatmaps parallel to each axis show the contributions of the 10 genes contributing most to the corresponding PC (**c, e, g**). Heatmaps show logFC for the 20 genes with the greatest absolute logFC values in the human differential gene expression analysis, and their orthologs in each mouse model, corresponding to the analyses illustrated in (**b**), (**d**), and (**f**), respectively. Mouse models are ordered left to right in order of increasing dissimilarity to the human disease, based on the Euclidian distance calculated from all principal components (*Supplementary file 13*). The rows (genes) are ordered by absolute logFC in the human comparison in descending order. n=3 for early and n=3 for late time point in each mouse model; n=21 Uncomplicated, n=8 HL, n=5 CM, n=12 CH. Full heatmaps for the expression of genes contributing most to the first two principal components in humans and each mouse model shown in *Figure 3—figure supplements 1–6*. The mouse model abbreviations are as follows: PbNK65 (*Plasmodium berghei* NK65), PbANKA (*P. berghei* ANKA), PcAS (*P. chabaudi* AS), Py17XL (*P. yoelii* 17XL), and Py17XNL (*P. yoelii* 17XNL).

The online version of this article includes the following figure supplement(s) for figure 3:

**Figure supplement 1.** Additional heatmaps for *Figure 3*.

**Figure supplement 2.** Additional heatmaps for *Figure 3*.

**Figure supplement 3.** Additional heatmaps for *Figure 3*.

**Figure supplement 4.** Additional heatmaps for *Figure 3*.

**Figure supplement 5.** Additional heatmaps for *Figure 3*.

**Figure supplement 6.** Additional heatmaps for *Figure 3*.

---

differentially expressed genes were not concordantly regulated (*Figure 3c*, *Supplementary file 9*). Amongst the most concordant genes were those encoding neutrophil granule proteins: Lactotransferrin (*LTF, Ltf*), Olfactomedin 4 (*OLFM4, Olfm4*), CD177 (CD177, *Cd177*), Matrix Metallopeptidase 8 (MMP8, *Mmp8*), Lipocalin 2 (LCN2, *Lcn2*), Matrix Metallopeptidase 9 (MMP9, *Mmp9*), and S100 Calcium Binding Protein A8 (S100A8, *S100a8*); but there was notable discordance of expression of genes encoding Arginase 1 (ARG1, *Arg1*), Cathepsin G (CTSG, *Ctsg*), Resistin (RETN, *Retn*), Vascular Cell Adhesion Molecule 1 (VCAM1, *Vcam1*), and Secreted Phosphoprotein 1 (SPP1, *Spp1*) (*Figure 3c*).

In the comparison of the mouse models with the human CM phenotype (*Figure 3d*), *P. yoelii* 17XL was again the mouse model with greatest similarity in gene expression changes, and GO analysis revealed that myeloid leucocyte activation and neutrophil degranulation were again the most enriched GO terms amongst the genes explaining the greatest variation between models (*Supplementary file 11*). The genes with concordant and discordant changes in expression between humans and mice were also similar to those in the HL comparison.

Findings were similar when we compared the changes in gene expression in the mouse models with those in children with UM versus children with the most severe phenotype where both CM and HL are present (CH) (*Lee et al., 2018b*). *P. yoelii* 17XL was placed closest to human CH in the PCA plot (*Figure 3g*), and the genes contributing most to PC1 and PC2 were again enriched in neutrophil degranulation and myeloid leukocyte activation GO terms (*Supplementary file 11*). The finding that neutrophil degranulation and myeloid leukocyte activation pathway genes account for the greatest variation between human SM phenotypes and the mouse models is consistent with increasing evidence that different aspects of neutrophil function could contribute to pathogenesis or protection from SM in both humans and some mouse models (*Lee et al., 2018b*; *Knackstedt et al., 2019*; *Georgiadou et al., 2021*; *Aitken et al., 2018*; *Feintuch et al., 2016*; *Sercundes et al., 2016*). Taken together, the comparisons between mouse models and these three SM phenotypes in Gambian children suggest that *P. yoelii* 17XL recapitulates the profile of the most prominent changes in gene expression associated with human SM phenotypes more closely than the other mouse models.

The relative frequency of different manifestations of *P. falciparum* SM varies across different geographic locations, influenced by the intensity of exposure to malaria, naturally acquired immunity, and age of individuals (*Wassmer et al., 2015*; *Okiro et al., 2009*). Changes in gene expression associated with the same disease manifestation may also vary between studies in different populations, under genetic and environmental influences, and due to technical differences in the methods used to

assess gene expression (*Driss et al., 2011*; *Wang et al., 2009*). Therefore, we investigated whether similar results would be obtained using data from an independent study conducted in Gabonese children with *P. falciparum* infection (*Boldt et al., 2019*).

In the study from which we obtained this data, Gabonese children with CM and CH (CM/CH) were not distinguished as separate phenotypes and were pooled into a single group for microarray analysis (see Materials and methods). Nevertheless, there was relatively high (78%) concordance of differentially expressed genes in the Gambian CH-UM and Gabonese CM/CH-UM comparisons (*Supplementary files 1D and 9*). Comparison of changes in gene expression between early and late stages of the mouse infections with those between Gabonese children with UM and CM/CH revealed that *P. yoelii* 17XL most closely recapitulated the differential expression seen in humans (*Figure 4*, *Supplementary files 12 and 7*). GO analysis confirmed that the innate immune response and leukocyte mediated immunity were the main drivers of variation between models, similar to the analysis in Gambian children (*Supplementary file 11*).

In contrast to the Gambian data set, where SA was rare (*Cunnington et al., 2013*), the SA phenotype was included in the Gabonese data set. Comparing the differential gene expression in the mouse models and those between UM and SA also identified that the changes in gene expression seen in *P. yoelii* 17XL were most similar to the differences seen in the Gabonese children (*Figure 4c*). The genes with highly concordant expression between SA and *P. yoelii* 17XL were prominently neutrophil related (*LTF*, *OLFM4*, *MMP9,* and *IL1R2*) (*Figure 4d*), GO analysis revealed that the main drivers of PC1 were slightly different to previous comparisons with prominence of immune response and type I interferon signaling pathways, whilst PC2 drivers were more similar to previous comparisons including leukocyte activation and neutrophil degranulation (*Supplementary file 11*). Type I interferon signaling, known for its immune-modulatory and anti-viral functions, has conflicting roles in human and mouse malaria (*Sebina and Haque, 2018*; *He et al., 2020*) dependent on timing and persistence of expression, which can be either advantageous (*Kempaiah et al., 2012*; *Krupka et al., 2012*; *Subramaniam et al., 2015*; *Yu et al., 2016*) or detrimental (*Feintuch et al., 2018*; *Capuccini et al., 2016*; *Spaulding et al., 2016*) to the host. The data from Gabonese children provide independent, cross-platform, comparison, and substantiate that the profile of gene expression associated with severe *P. yoelii* 17XL infection is most similar to those in the major human SM phenotypes.

The lack of suitable publicly available gene expression data sets from mice infected with the range of malaria parasites used in our analyses precluded exploration of whether similar results would be obtained from mouse infection experiments conducted in different laboratories or using different platforms to assess gene expression. However, a comparison of differential expression in published microarray data from blood of early and late-stage *P. chabaudi* AS infections (*Lin et al., 2017*) showed high (86%) concordance with the differentially expressed genes in equivalent analysis in our mouse RNA-Seq data set (see Materials and methods; *Supplementary file 2*, *Supplementary file 9*, *Supplementary file 16*), suggesting that generalizability is likely.

## Comparative transcriptomic results are consistent with pathophysiology

The profile of changes in gene expression associated with HL, CM, and SA, the three most common manifestations of SM in children, were all better recapitulated by the changes in gene expression in *P. yoelii* 17XL than any other mouse model. However, this model is not widely used to study the pathogenesis of these specific SM syndromes, so we sought to determine whether *P. yoelii* 17XL does reproduce the pathophysiological features of these infections. Blood lactate levels have rarely been reported in mouse malaria models, so we systematically measured lactate concentrations at early and late stages of infection in all five mouse models (*Figure 5a*, *Figure 5—source data 1*). Small differences, if any, were noticed at the uncomplicated stage early in infection, while at maximum severity *P. yoelii* 17XL and *P. berghei* NK65 infected mice developed dramatic hyperlactatemia with concentrations similar to the maximum values seen in human HL (*Lee et al., 2018b*).

*P. yoelii* 17XL also reproduced the changes in gene expression associated with human SA better than other mouse models. Human SA is often associated with very high parasite biomass (*Cunnington et al., 2013*) and *P. yoelii* 17XL achieves much higher parasite load than other mouse models (*Figure 1*) as well as causing rapid and profound anemia (*Couper et al., 2007*; *Totino et al., 2010*; *Figure 5b*).

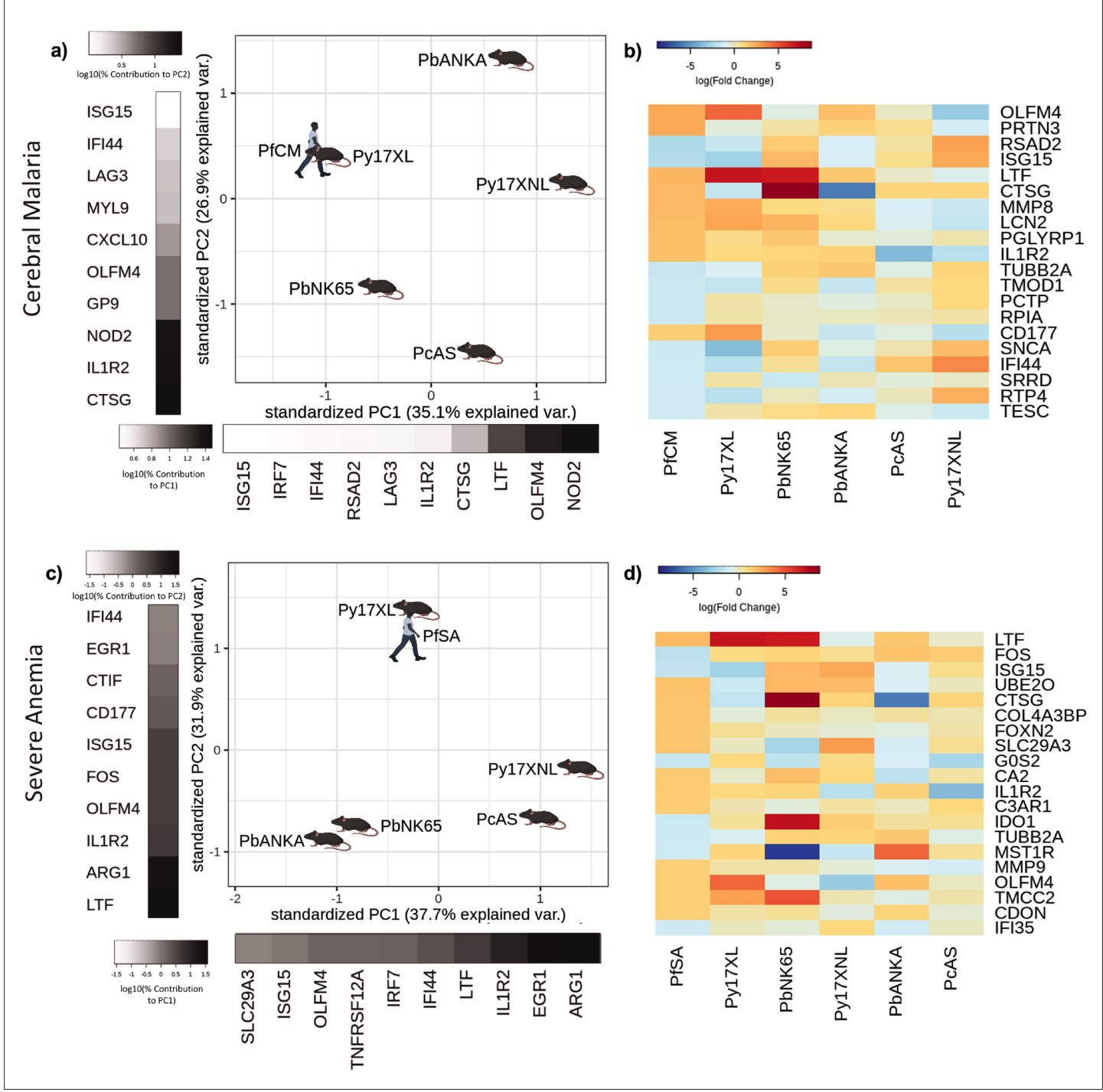

**Figure 4.** Comparison of host differential gene expression in two severe malaria phenotypes in Gabonese Children and five mouse malaria models. (**a, c**) Principal component analysis (PCA) plots generated using rank-normalized log-fold change values from the human and mouse differential expression analyses. Only genes with 1:1 mouse and human orthologs and with absolute logFC value greater than 1 in the corresponding human comparison were included. Comparison of changes in gene expression in the mouse models with those in human cerebral malaria (PfCM) (**a**) and severe anemia (PfSA) (**c**). The percentage of the total variation explained by principal components 1 and 2 are shown in the axis labels. Grayscale heatmaps parallel to each axis show the contributions of the 10 genes contributing most to the corresponding PC (**b, d**). Heatmaps show logFC for the 20 genes with the greatest absolute log-fold change values in the human differential gene expression analysis, and their orthologs in each mouse model, corresponding to the analyses illustrated in (**a**) and (**c**). Mouse models are ordered left to right in order of increasing dissimilarity to the human disease, based on the Euclidian distance calculated from all principal components (***Supplementary file 13***). The rows (genes) are ordered by absolute log-fold change in the human comparison in descending order. n=3 for early and n=3 for late time point in each mouse model; n=5 pooled samples uncomplicated (UM), n=5 pooled

*Figure 4 continued on next page*

*Figure 4 continued*

samples CM, n=5 pooled samples SA (each pool contained RNA from four individuals with the same phenotype). Full heatmaps for the expression of genes contributing most to the first two principal components in humans and each mouse model shown in *Figure 4—figure supplements 1–4*. The mouse model abbreviations are as follows: PbNK65 (*Plasmodium berghei* NK65), PbANKA (*P. berghei* ANKA), PcAS (*P. chabaudi* AS), Py17XL (*P. yoelii* 17XL), and Py17XNL (*P. yoelii* 17XNL).

The online version of this article includes the following figure supplement(s) for figure 4:

**Figure supplement 1.** Additional heatmaps for *Figure 4*.

**Figure supplement 2.** Additional heatmaps for *Figure 4*.

**Figure supplement 3.** Additional heatmaps for *Figure 4*.

**Figure supplement 4.** Additional heatmaps for *Figure 4*.

*P. yoelii* 17XL also showed the greatest transcriptional similarity to the pattern of changes in whole blood gene expression associated with human CM. *P. yoelii* 17XL was originally described as a virulent clone causing CM-like pathology (*Yoeli and Hargreaves, 1974*), but it has subsequently been replaced by *P. berghei* ANKA as the most commonly used model of experimental CM. Since one of the key pathological mechanisms leading to death in pediatric CM is brain swelling due to extravascular fluid leak (*Moxon et al., 2020*), we examined the presence of extravascular fibrinogen (*Georgiadou et al., 2021*) as an indicator of vascular leak in the brains of both *P. berghei* ANKA and *P. yoelii* 17XL infected mice compared to uninfected mice (*Figure 5c*). We found that brains from both infections had areas that stained positively for perivascular fibrinogen (indicative of vascular leak), while additionally some of the vessels from *P. yoelii* 17XL infected mice showed strong intravascular staining, suggestive of microthrombus formation (*Figure 5c, iv* ), another mechanism that has been implicated in human CM (*Moxon et al., 2020*; *Georgiadou et al., 2021*).

## Discussion

Mice are the most cost effective and widely used model organism for studying many human diseases (*Stuart et al., 2003*; *Zheng-Bradley et al., 2010*). Nevertheless, mice are distant evolutionarily and differ substantially from humans in many ways (*Mestas and Hughes, 2004*; *Liao and Zhang, 2008*). Disease models in mice often involve artificial induction of disease, which may reduce complexity and aid reproducibility, but might also limit their translational relevance. Therapeutic interventions that work in mice often fail when used in human clinical trials (*Bugelski and Martin, 2012*; *Hünig, 2012*). As a result, the usefulness of mice in some areas of translational research is debated (*Shay et al., 2013*; *Seok et al., 2013*). Recently, concerted efforts have been made to improve both scientific and ethical aspects of the use of animals in biomedical research, with emphasis on the principles of replacement, reduction, and refinement (the "3Rs"), and improving reproducibility through better experimental design and standardized reporting guidelines (*Percie du Sert et al., 2020*). Despite this, there has been little parallel effort made to assess or improve the relevance of animal models in translational research, and approaches that would improve translation from mice to humans are needed (*Normand et al., 2018*).

In malaria research, mouse models are widely used but their relevance to human disease is contentious (*Craig et al., 2012*). Here, we objectively assessed the biological processes occurring in blood in some of the most commonly used mouse models of malaria to examine their similarity to human malaria, using a comparative transcriptomic approach. The five rodent malaria parasites, we used led to the development of distinct disease trajectories and clinical features. Whilst no rodent malaria parasites induced changes in gene expression which fully recapitulated those in human malaria, at an early stage of infection, the rodent malaria parasites induced relatively similar transcriptional host responses to each other, with at least a broad overall similarity to that seen in a large study of UM in African children, and CHMI in malaria naïve adults. However, when we investigated the similarity of the changes in gene expression associated with different SM manifestations, we saw that there was more heterogeneity, and the concordance and discordance of expression of individual genes varied more between each mouse model and each phenotype. One of the greatest sources of variation between the mouse models was in the myeloid cell response, particularly neutrophil response, associated with SM manifestations.

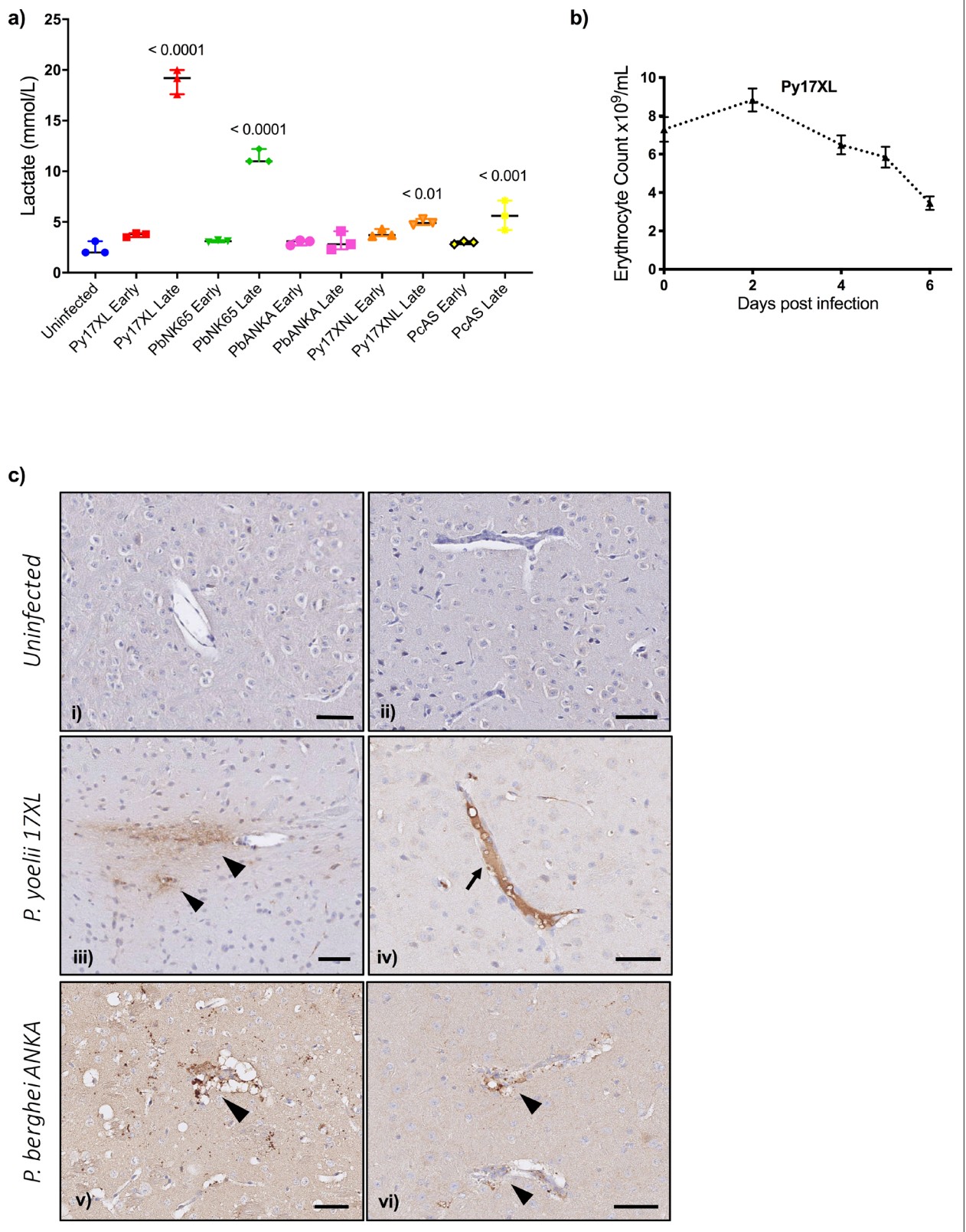

**Figure 5.** Pathophysiological features of rodent malaria infections. (**a**) Lactate concentration in blood (mmol/L) in mice, uninfected, or at the early or late stage of each malaria parasite infection (n=3 for each infection time point). Error bars show median with range, One-way ANOVA p-value<0.0001, p-values for post hoc Dunnett's multiple comparisons against uninfected mice are shown within the plot. (**b**) Erythrocyte counts from *Plasmodium yoelii 17XL* infected mice, n=9, representative of three experiments, repeated measures ANOVA p-value<0.01. (**c**) Representative histological specimens

*Figure 5 continued on next page*

Figure 5 continued

of brain with fibrinogen staining to identify vascular leak in mice uninfected (**i, ii**), infected with *P. yoelii 17XL* (**iii, iv**), and infected *P. berghei ANKA* (**v, vi**) collected at the late stage (humane endpoint) of infection. Arrowheads identify extravascular fibrinogen indicating leak from the vasculature. Arrow points to strong intravascular fibrinogen staining (**iv**) suggestive of microthrombus. Representative images from analysis of uninfected mouse brains n=3; *P. yoelii* 17XL-infected mouse brains n=5; *P. berghei ANKA*-infected mouse brains n=4; Scale bar: 50 µm. Eight-week-old wild-type female C57BL/6J mice were used in all experiments. Individual mouse lactate measurements and erythrocyte counts shown in *Figure 5—source data 1*.

The online version of this article includes the following source data for figure 5:

**Source data 1.** Individual mouse lactate measurements and erythrocyte counts.

An important implication of our findings is that the selection of the most appropriate mouse model for investigation of a particular mechanism of interest should not be made solely on the similarity of clinical phenotype in humans and mice. We propose that it should also be based on the degree of concordance of expression of genes associated with the mechanism of interest. Failure to consider the similarities and differences in biological processes indicated by gene expression could lead to experiments targeting pathways that are not involved in the host response to a particular mouse malaria parasite, making the experiments futile, unethical, and potentially leading to erroneous conclusions.

We identified that the pattern of changes in gene expression between early and late stages of *P. yoelii* 17XL infected mice showed the greatest similarity to the differences in gene expression between human UM and each of HL, CM, CH, and SA, suggesting that this model might be most representative of the profile of changes in host response induced by human SM. This mouse model not only develops a very high parasite load, but our data suggest lethality at 5–7 days post-infection is part of a multisystem disorder, accompanied by extreme hyperlactatemia at levels similar to those seen in human HL and CH. Until now, the lack of a rodent model to study malaria-induced hyperlactatemia has held back understanding of the mechanisms causing such high levels of lactate and how these relate to the increased risk of death in patients with malaria. *P. yoelii* 17XL infection of C57BL/6J mice is an attractive model for further translational research on this SM phenotype.

In the brains of *P. yoelii* 17XL infected mice, we identified extravascular fibrinogen leak. This suggests that these mice may be in the process of developing a neurological syndrome at the time they reach the humane endpoint and may explain why this model showed transcriptional similarity to human CM. The transcriptional similarity of Py17XL to the human SA phenotype is consistent with the SA and high parasite load which occurs in this infection.

Despite Py17XL appearing to have the closest overall transcriptional similarity to human SM syndromes, we identified many genes with discordant expression, and Py17XL infection may not recapitulate all pathophysiological features of human SM. The maximum parasitemia (~80%) seen in Py17XL infection is much higher than that typically seen in human SM (~10%) (*Cunnington et al., 2013*), although the sequestration of *P. falciparum*-infected red cells in human SM means that total parasite load may be several fold-higher than indicated by peripheral blood parasitemia (*Cunnington et al., 2013*).

Our study provides important insights into the translational relevance of commonly used mouse models of malaria, and more generally highlights the importance of considering relevance in addition to the 3Rs and reproducibility when planning any animal experiments. Our data are provided as a resource for researchers to help them to determine the concordance of gene expression between mouse malaria models and human disease, and we have identified an attractive mouse model for further translational studies on malarial hyperlactatemia. A strength of analyzing the blood transcriptome is that it represents the systemic host response to infection, capturing both the direct influence of an infectious agent on blood leukocytes, and the response of blood leukocytes to mediators released into the circulation by cells in other organs. However, the blood transcriptome cannot assess the concordance of processes occurring within specific organs that do not produce changes in gene expression of circulating leukocytes, and our data should not be used to prevent testing of reasonable hypotheses about such tissue-specific interactions. Reassuringly, our findings were broadly consistent when we performed comparisons across independent studies conducted in different locations and using different transcriptomic methods. Stronger and more generalizable conclusions, and more nuanced approaches to analysis may be possible if future studies add to the data we have collected, with larger numbers of mice and greater sequencing depth. Future work should also assess other commonly used mouse malaria models, using additional common mouse strains (including Balb/c and DBA/2) and outbred mice, both sexes, additional parasite strains, and mosquito-transmitted infections.

# Materials and methods

**Key resources table**

| Reagent type (species) or resource | Designation | Source or reference | Identifiers | Additional information |
|---|---|---|---|---|
| Strain, strain background *Plasmodium berghei* | NK65 | https://doi.org/10.4049/jimmunol.0904019 | | |
| Strain, strain background *P. berghei* | ANKA | https://doi.org/10.4049/jimmunol.1100241 | | |
| Strain, strain background *P. yoelii* | 17XL | https://doi.org/10.1371/journal.ppat.1000004 | | |
| Strain, strain background *P. yoelii* | 17XNL | doi:10.1002/eji.201546018 | | |
| Strain, strain background *P. chabaudi* | AS | https://doi.org/10.1111/j.1365-3024.2012.01366.x | | |
| Strain, strain background (*Mus musculus*, female) | C57BL/6J | Charles River Laboratories | | |
| Antibody | Anti-Fibrinogen antibody Rabbit polyclonal antibody | Abcam | ab34269 RRID:AB_732367 | (1:100) |
| Antibody | Alexa Fluor 488 anti-mouse/human CD11b Clone M1/70 Rat monoclonal antibody | BioLegend | (101217) RRID:AB_389305 | (1:300) |
| Antibody | APC anti-mouse Ly-6G Clone 1 A8 Rat monoclonal antibody | BioLegend | (127614) RRID:AB_2227348 | (1:300) |
| Antibody | PE anti-mouse CD19 Clone 6D5 Rat monoclonal antibody | BioLegend | (115508) RRID:AB_313643 | (1:300) |
| Antibody | Brilliant Violet 421 anti-mouse CD4 Clone GK1.5 Rat monoclonal antibody | BioLegend | (100443) RRID:AB_2562557 | (1:200) |
| Antibody | Alexa Fluor 700 anti-mouse CD8a Clone 53–6.7 Rat monoclonal antibody | BioLegend | (100730) RRID:AB_493703 | (1:200) |
| Antibody | Brilliant Violet 650 anti-mouse CD3 Clone 17 A2 Rat monoclonal antibody | BioLegend | (100229) RRID:AB_11204249 | (2:100) |
| Antibody | Alexa Fluor 488 anti-mouse CD4 Clone GK1.5 Rat monoclonal antibody | BioLegend | (100425) RRID:AB_493520 | (1:200) |
| Antibody | APC anti-mouse CD4 Clone GK1.5 Rat monoclonal antibody | BioLegend | (100411) RRID:AB_312696 | (1:200) |
| Antibody | PE anti-mouse CD4 Clone GK1.5 Rat monoclonal antibody | BioLegend | (100407) RRID:AB_312692 | (1:200) |
| Antibody | Alexa Fluor 700 anti-mouse CD4 Clone GK1.5 Rat monoclonal antibody | BioLegend | (100429) RRID:AB_493698 | (1:200) |
| Antibody | Brilliant Violet 650 anti-mouse CD4 Clone GK1.5 Rat monoclonal antibody | BioLegend | (100545) RRID:AB_11126142 | (1:200) |
| Commercial assay or kit | PAXgene Blood RNA Kit | QIAGEN | Cat. No./ID: 762174 | |
| Commercial assay or kit | Agilent RNA 6000 Nano Kit | Agilent | 5067-1511 | |

*Continued on next page*

Continued

| Reagent type (species) or resource | Designation | Source or reference | Identifiers | Additional information |
|---|---|---|---|---|
| Software, algorithm | R | https://www.R-project.org/ | R 3.5.1 RRID:SCR_001905 | |
| Software, algorithm | STAR | DOI:10.1093/bioinformatics/bts635 | 2.5.4b RRID:SCR_004463 | |
| Software, algorithm | Python Package: HTSeq | DOI:10.1093/bioinformatics/btu638 | 1.99.2 RRID:SCR_005514 | |
| Software, algorithm | GraphPad PRISM | https://www.graphpad.com | GraphPad Prism 8 RRID:SCR_002798 | |

## Experimental design

We compared the whole blood transcriptome changes associated with SM in mice and humans to identify concordant and discordant patterns of gene expression, and to identify which mouse models show the most similar changes to those seen in humans.

We chose to compare the changes in gene expression between human UM and SM categories with those seen between early and late mouse infections, assuming that mice early in infection (when the first symptoms occur) represent UM while mice at the peak of severity symptoms (or humane endpoints) represent SM. Human data were obtained from published data sets from our group (*Lee et al., 2018b*) and others (*Idaghdour et al., 2012*; *Boldt et al., 2019*; *Milne et al., 2021*) while mouse data were generated specifically for this experiment and extracted from limited publicly available data (*Lin et al., 2017*).

## Animals and procedures

Eight-week-old wild-type female C57BL/6J mice were obtained from Charles River Laboratories. All mice were specified pathogen-free, housed in groups of five in individually ventilated cages, and allowed free access to food and water. All protocols and procedures were approved by Imperial College Animal Welfare and Ethical Review Board, following Laboratory Animal Science Association good practice guidance. Mice were acclimatized to the animal facility for 1 week before any experimental procedures.

Parasites (*P. berghei* ANKA [lethal], *P. berghei* NK65 [lethal], *P. yoelii* 17XL [lethal], *P. yoelii* 17XNL [non-lethal], and *P. chabaudi* AS [non-lethal]) were a kind gift from Professor Eleanor Riley and had been serially blood passaged through C57BL/6 mice (*Findlay et al., 2010*; *Villegas-Mendez et al., 2011*; *Couper et al., 2008*; *Stegmann et al., 2015*; *Toscano et al., 2012*). Parasites stocks were stored in Alsever's solution with 10% glycerol (mixed at 1:2 ratio) and were defrosted and diluted (depending on parasitemia of the frozen stock) to infect a passage mouse. The passage mouse infection was then closely monitored until healthy parasites were observed in a blood smear and parasitemia reached at least 2%. Blood was collected, before parasitemia reached 5%, by aseptic cardiac puncture under non-recovery isoflurane anesthesia, and diluted in sterile phosphate-buffered saline to achieve desired concentration. Experimental mice were infected with $10^5$ live parasites by intraperitoneal injection. Fifty mice were randomly allocated to be infected in groups of 10 with each parasite strain and then segregated into two cages of five mice each per parasite strain. Ten control uninfected mice were used for weight-gain comparisons.

The weight and physical condition of each mouse were monitored throughout the course of each infection (*Figure 1—figure supplement 1*, *Figure 1—source data 1*). Change in weight was calculated as a percentage of baseline weight measured prior to infection. For *P. berghei* ANKA infection, which causes ECM, additional neurological monitoring was performed using the Rapid Murine Coma and Behaviour Scale (RMCBS) (*Carroll et al., 2010*), which includes assessment of gait, motor performance, balance, limb strength, body position, touch escape, pina reflex, foot withdrawal reflex, aggression, and grooming. Due to the need for different intensity and nature of monitoring in each infection to ensure animal welfare, blinding to infection group was considered inappropriate.

The early time point was defined as the first time at which mice manifested any signs of ill health, including any reduction in activity, ruffled fur or, weight loss. The late time point was defined as the

humane endpoint for each lethal parasite strain (*Figure 1—figure supplement 1*, *Supplementary file 17*), or a time point chosen to be just before the expected day of maximum severity of non-lethal infections (to avoid sampling mice which were starting to recover).

Tail capillary blood was used to prepare blood smears for analysis of parasitemia and lactate measurement using the Lactate Pro 2 (HAB direct) lactate meter. Parasitemia was quantified by microscopy of thin blood smears stained with 10% Giemsa and examined at 100× magnification with a Miller Square reticle. Erythrocyte counts were determined using a Z2 Coulter particle counter (Beckman Coulter). When mice were euthanized, heparinized blood was collected by cardiac puncture under non-recovery isoflurane anesthesia, and an aliquot of 300–500 µl was immediately mixed at 1:2.76 volume ratio with fluid from a PAXgene Blood RNA Tube (QIAGEN), whilst the remainder was stored on ice for flow cytometry analysis. Brains were collected from *P. yoelii* 17XL and *P. berghei* ANKA infected mice and fixed in 4% paraformaldehyde for 48 hr before being processed. Brains were then paraffin embedded, cut, and stained with antibody against fibrinogen (ab34269 1:100, Abcam, UK) using a Roche automated staining system. Digitized images were taken at 40× magnification (LEICA SCN400, Leica Microsystems UK) at IQPath (Institute of Neurology, University College, London, UK). Images were then viewed and examined with Aperio ImageScope software (Leica Biosystems Imaging, Inc).

## Flow cytometry

The proportions of major leukocyte subpopulations in mouse blood were determined by flow cytometry using specific cell-surface marker antibodies. Approximately 50 µl of whole blood was mixed with 2 ml ammonium chloride red-cell lysis buffer for 5 min at room temperature, then samples were centrifuged and washed in flow cytometry buffer and centrifuged again. Resultant cell pellets were resuspended in 50 µl of antibody cocktail (all antibodies from BioLegend, Key resources table) for 30 min before further washing and fixation in 2% paraformaldehyde. Flow cytometry was performed using a BD LSR Fortessa machine. BD FACSDiva software was used to collect the data and analysis was conducted using FlowJo v10 (TreeStar Inc), gating on single leukocytes before identification of major cell populations according to their surface marker staining (*Supplementary file 1E*). Leucocyte proportions for early and late timepoints within each infection are presented in *Supplementary file 1F*.

## Comparison of cell type proportions between species

The proportions of lymphocytes, neutrophils, and monocytes, measured by haematology analyser, in human malaria subjects (*Lee et al., 2018b*) were compared to the proportions of lymphocytes (sum of the B-lymphocyte and CD4+ T-lymphocyte and CD8+ T-lymphocyte), neutrophils, and monocytes, measured by flow cytometry, in the mouse RNASeq data set.

## RNA isolation from mouse blood

RNA extraction was performed using the PAXgene Blood RNA Kit (QIAGEN ) according to the manufacturers' instructions (*Meyer et al., 2016*). After the isolation of the RNA, Nanodrop ND-1000 Spectrophotometer (LabTech) was used to obtain the ratio of absorbance at 260 nm and 280 nm (260/280) which is used to assess the purity of RNA (or DNA). Values of ~2 are generally accepted as pure for RNA. RNA integrity was assessed using Agilent RNA 6000 Nano Kit (Agilent), used according to the manufacturers' instructions with the Agilent 2100 Bioanalyzer (Agilent), and all traces were inspected visually for evidence of RNA degradation because the RNA Integrity Number calculation can be misleading when host and parasite RNA are both present in significant quantities (*Lee et al., 2018b*).

For the RNA sequencing analysis, six samples were selected from each infection (three from the early time point and three from the late time point), along with three uninfected controls. Samples were selected based on the RNA quality (260/280 ratio and Agilent 2100 Bioanalyzer traces). If more than three samples for each infection and time point were of sufficient quality, we selected the three with most similar clinical score and parasitemia levels within each group.

## Dual-RNA sequencing

Library preparation and sequencing to generate the mouse RNA-Seq data was performed at the Exeter University sequencing service. Libraries were prepared from 1 µg of total RNA with the use of ScriptSeq v2 RNA-Seq Library Preparation Kit (Illumina) and the Globin-Zero Gold Kit (Epicentre) to

remove globin mRNA and ribosomal RNA. Prepared strand-specific libraries were sequenced using the 2×125 bp protocol on an Illumina HiSeq 2500 instrument.

## Gene annotations

Human reference genome (hg38) was obtained from UCSC genome browser (http://genome.ucsc.edu/), mouse reference genome (mm10) was obtained from UCSC genome browser (http://genome.ucsc.edu/). Human gene annotation was obtained from GENCODE (release 22) (http://gencode-genes.org/releases/), mouse gene annotation was obtained from GENCODE (release M16) (http://gencodegenes.org/releases/). The *Plasmodium* (*P. berghei, P. chabaudi,* and *P. yoelii*) genomes were obtained from PlasmoDB (release 24) (*Aurrecoechea et al., 2009*).

## Mouse RNA-Seq quality control, mapping, and quantification

Quality control was carried out using fastqc (*Andrew, 2010*) and fastqscreen (*Wingett and Andrews, 2018*). Adapters were trimmed using cutadapt (*Martin, 2011*). The read 1 (R1, -a) adapter is AGAT CGGAAGAGCACACGTCT, and the read 2 (R2, -A) adapter is AGATCGGAAGAGCGTCGTGTAGGG AAAGAGTGT.

The trimmed reads were then mapped to the combined genomic index containing both mouse and the appropriate *Plasmodium* genome using the splice-aware STAR aligner (*Dobin et al., 2013*). Reads were extracted from the output BAM file to separate parasite-mapped reads from mouse-mapped reads. Reads mapping to both genomes were counted for each sample and removed. BAM files were sorted, read groups replaced with a single new read group, and all reads assigned to it. HTSeq-count (*Anders et al., 2015*) was used to count the reads mapped to exons with the parameter "-m union." Only uniquely mapping reads were counted.

## Confirmation of parasites species and strain

We confirmed the purity and identity of parasite strains by the unique mapping of non-mouse RNA reads from each infection to the respective parasite species genome using fastQ-screen. We then confirmed the presence of expected polymorphisms distinguishing between *P. yoelii* strains and between *P. berghei* strains by using the RNA-Seq data to identify distinctive single-nucleotide poly-morphisms (SNPs).

For SNP identification, *P. yoelii* samples were mapped to the *P. yoelii* 17X (also called Py17XNL) genome and *P. berghei* samples were mapped to the PbANKA genome (extracted from PlasmoDB) using bwa-mem (*Li, 2013*), and sorted using samtools (*Li et al., 2009*).

The phenotypic differences between the two strains of *P. yoelii* are due to an SNP in the PyEBL gene (PY17X_1337400), changing a T (in the Py17XNL reference) to an A (in Py17XL) (*Otsuki et al., 2009*) at chromosomal location 1,704,423. The presence of the reference sequence in the *P. yoelii* 17XNL infection samples and mismatch at this location in *P. yoelii* 17XL infection samples was confirmed using Integrative Genomics Viewer (IGV) (*Robinson et al., 2011*; *Supplementary file 1G*).

In contrast to the *P. yoelii* strains, there are many SNPs that distinguish *P. berghei* ANKA from *P. berghei* NK65 (*Akkaya et al., 2020*). Four SNPs (in genes PbANKA_1331700, PbANKA_0515200.1, PbANKA_1414600, and PbANKA_1222100.1) were examined in IGV using the *P. berghei* ANKA genome as a reference, confirming the absence of mismatch in the *P. berghei* ANKA infection samples and the presence of the expected mismatches in the *P. berghei* NK65 infection samples (*Supplementary file 1H*).

## Mouse differential gene expression analysis

The Ensembl gene ID versions were matched to their MGI gene symbols and Entrez IDs using biomaRt (annotation used: http://jul2018.archive.ensembl.org, mmusculus_gene_ensembl) (*Durinck et al., 2009*; *Durinck et al., 2005*). Genes for which this information was not available were excluded from the analysis. Of these, only genes with raw expression values of greater than 5 in at least three samples were taken forward. The raw expression counts can be found in *Supplementary file 18*.

The differential gene expression analysis was then performed using the R package edgeR. Raw read counts of each data set were normalized using a trimmed mean of M-values (TMM), which considers the library size and the RNA composition of the input data.

In order to account for variation between samples in the proportions of the major blood leukocyte populations (neutrophil, monocyte, CD4 T cell, and CD8 T cell), we used their proportions estimated by flow cytometry (*Supplementary file 19*) as covariates in edgeR, adjusting for their effect on whole blood gene expression. B cells were excluded from the design matrix of the differential expression analysis due to the proportions totaling 100%. Thus, the design matrix (with the intercept set to 0) consisted of each sample's disease type (the mouse model plus if the sample was early or late in infection, i.e., *P. yoelii* 17XL_late) with the cell type proportions as covariates. Results of the differential expression analyses are presented in *Supplementary file 12*. Metadata matching each sample to their phenotype can be found in *Supplementary file 20*.

## Analysis of the human RNA-Seq data set

For the comparison with RNA-Seq data from human hosts, data from our previously published Gambian child cohort were used (*Lee et al., 2018b*). This data set can be found in the ArrayExpress database (https://www.ebi.ac.uk/arrayexpress) using the accession number E-MTAB-6413 and metadata are also presented in *Supplementary file 21*. Differential expression analysis and adjustment for cell mixture were performed as previously described using CellCode and EdgeR (*Lee et al., 2018b*). Lists of differentially expressed genes are available in *Supplementary file 14*.

## Analysis of microarray data sets

Expression values for three human microarray data sets were extracted from the GEO database (*Idaghdour et al., 2012*; *Boldt et al., 2019*; *Milne et al., 2021*; *Supplementary file 1B*). For the Boldt et al. study, background correction, normalization, and batch correction were performed on the raw expression values using the methods given in *Supplementary file 1B*. For the Idaghdour et al. study, the data was downloaded as pre-normalized expression values. For the Milne et al. CHMI data set, the raw cel files were downloaded from the GEO database.

For all three data sets, CellCODE (*Chikina et al., 2015*) was used to estimate the proportions of the major blood leukocyte subpopulations (neutrophils, monocytes, CD4 T cells, CD8 T cells, and B cells) in each of the samples for all three microarray data sets. This was based on reference gene expression profiles, Allantaz et al. GEO Accession: GSE28490 (*Allantaz et al., 2012*) the full signature data set derived from Allantaz et al., not just those used for these data sets, can be found in *Supplementary file 22*. Surrogate proportion variables for each leukocyte subpopulation were then used as covariates in differential gene expression pairwise analyses in Limma (*Smyth, 2005*; *Supplementary file 1B*).

One sample (GSM848487) was removed from the Idaghdour et al. data set because the age of the subject was not available. The original study sampled a population with wide age range from different locations, so following the approach in the original study, differential expression analysis included age, location (Zinvie or Cotonou), and hemoglobin genotype (AA, AS, or AC), in addition to the leukocyte subpopulation surrogate proportion variables estimated from CellCODE (*Chikina et al., 2015*), as covariates for the pairwise differential expression analysis conducted using Limma.

For the Milne et al. CHMI data set, the samples from a Malaysian individual (who may have had previous malaria) were removed and paired samples from the remaining 14 subjects at day –1 (before infection) and day of diagnosis were used.

The lists of differentially expressed genes for these data sets are available in *Supplementary file 6*; *Supplementary file 7*, *Supplementary file 10*. For each microarray data set, differential expression analysis was also performed without adjustment for cell type proportions and are given in *Supplementary files 3-5*.

Additionally, a microarray mouse *P. chabaudi* AS data set was also extracted from the GEO database (*Lin et al., 2017*) and used to perform differential expression analysis between the early and late infection stages (*Supplementary file 1B*, no cell type mixture adjustment was performed). The results were compared to those of the early versus late stages of infection with *P. chabaudi* AS in the RNA-Seq data set (without adjustment for cell type proportions, *Supplementary file 2*, *Supplementary file 7*). Genes with an absolute logFC value of at least 1 in both comparisons were used for discordance-concordance analysis as described below.

### Identification of orthologous genes

A text file of all the orthologous (Ensembl 52) *Homo sapiens* (NCBI36) and *Mus musculus* genes was extracted from the Ensembl database and used as a reference (*Supplementary file 8*).

### Comparative transcriptomics using principal component analysis

To use as much information as possible about changes in gene expression between conditions in human and mouse malaria data sets of varying size, we did not impose a p-value threshold but began by selecting all genes in the human differential expression analyses with absolute log-fold change greater than 1. We then selected those with 1:1 orthologs in mice, and used these genes for subsequent comparisons with gene expression in mice. There were no cutoffs applied based on the differences in expression between early and late-stage infection in mice. Therefore, our analyses assess the extent to which changes in mouse gene expression recapitulate those in humans, but do not address the reciprocal question of how well human gene expression recapitulates that in mice.

To compare patterns of gene expression associated with pathogenesis between species, without undue influence of species-specific variation in the baseline- or inducible-expression of each gene, we focused further analysis on the contrasts between comparable pairs of human and pairs of mouse infection states. Both human microarray UM versus healthy results were compared to the mouse early stage infection versus uninfected control results.

The human RNA-Seq (*Lee et al., 2018b*) CM versus UM, HL versus UM, CH versus UM, and microarray (*Boldt et al., 2019*) CM versus UM and SA versus UM results were compared to mouse late stage versus early stage of infection results for each mouse model.

To allow comparison of the relative magnitudes of changes in gene expression between the human and mouse models, we developed a rank-based analysis of the changes in expression in each human and mouse pairwise comparison. Genes were ranked in descending order of absolute log-fold change, with ties given the same minimum rank. Each gene was then assigned a value of 100 divided by rank, which was then multiplied by the sign of the original log-fold change. For example, if the original log-fold change was negative, the rank-standardized value would then be multiplied by –1. This approach means that the genes with greatest difference in expression between the conditions of interest within-species have the biggest effect on the comparative transcriptomic analysis between species. These values are presented in *Supplementary file 23* and were used as the input for subsequent PCA to highlight the differences and similarities between the mouse models and human disease comparisons in low-dimensional space. The PCAs were performed using the R-core function Prcomp() with default parameters and visualized using functions from the ggbiplot (*Vu, 2021*) and ggimage packages. The 10 genes that contributed the most to principal components 1 and 2 (a subset of those given in *Supplementary file 24*) were collected using the factoextra (*Kassambara and Mundt, 2021*) and FactoMineR (*Lê et al., 2008*) packages, specifically the PCA() function, with scale.unit set to FALSE to correspond to the default parameters of the Prcomp() function.

### Discordance-concordance analysis

The percentage of concordantly and discordantly expressed genes in comparisons between mouse and human were calculated based on the log-fold change values of orthologous genes with an absolute logFC value greater than 1 in the human differential expression analysis. In comparisons between human data sets, assessment was based on genes detected in both data sets with an absolute log-fold change value greater than 1 in the human RNA-Seq data set. Gene expression was considered concordant if the direction of change in expression was the same between the comparator groups, and discordant if the direction of change in expression was opposite.

### Gene ontology analysis

Lists of genes contributing greater than or equal to 0.1% to PC1 and/or PC2 were also extracted (*Supplementary file 24*). These were used as the genes of interest for GO term enrichment analysis performed using the goana.DGELRT() function (Package: Limma) (*Smyth, 2005*). The list of all the 1:1 orthologs used as the input for the PCA were used as the background gene lists (*Supplementary file 25*). Human gene IDs were fed to the GO term enrichment analysis. For each comparison in each data set, the Euclidean distances (*Supplementary file 13*) between each of the mouse models and the human data were calculated using standardized log-fold change values and the R-core dist() function.

## Heatmaps

The 20 genes with the greatest absolute log-fold change value in each human disease comparison were used to construct illustrative heatmaps using the heatmap.2() function from the R package gplots (*Gregory et al., 2021*).

The log(FC) values of all genes or the top 10 contributing the most to PC1 or PC2 of the PCA plots were extracted for the human and each mouse comparison and used to generate additional heatmaps (*Figure 2—figure supplements 1–4*, *Figure 3—figure supplements 1–6*, *Figure 4—figure supplements 1–4*). The heatmap.2() function was used to generate these plots. The sample rows of this plot are ordered according to increasing Euclidean distance from the human.

## Statistical tests

GraphPad Prism 8 (GraphPad Software) was used for statistical analyses of lactate concentration in the different mouse models and erythrocyte counts from *P. yoelii 17XL* infected mice. One-way ANOVA test was used to compare the lactate concentration in mice uninfected or infected at different time points and post hoc Dunnett's test for multiple comparisons. One-way ANOVA for repeated measures was used to analyze erythrocyte counts from *P. yoelii* 17XL infected mice. All tests were two-sided using a significance threshold of 5%.

## Acknowledgements

The author thank Professor Eleanor Riley for kind donation of the parasites. This work was supported by the UK MRC and the UK Department for International Development (DFID) under the MRC/DFID Concordat agreement and is also part of the EDCTP2 program supported by the European Union (MR/L006529/1 to AJC), Imperial College Dean's EPSRC Studentship (to CD), Imperial College-Wellcome Trust Institutional Strategic Support Fund (to AG), Sir Henry Wellcome Fellowship (206508/Z/17/Z to MK), and the analysis of patient data were supported by the NIHR Imperial Biomedical Research Centre. Exeter Sequencing Service was supported by an MRC Clinical Infrastructure award (MR/M008924/1), the Wellcome Trust Institutional Strategic Support Fund (WT097835MF), a Wellcome Trust Multi-User Equipment Award (WT101650MA), and a Biotechnology and Biological Sciences Research Council Longer and Larger award (BB/K003240/1).

## Additional information

### Funding

| Funder | Grant reference number | Author |
|---|---|---|
| Imperial College London | Imperial College-Wellcome Trust Institutional Strategic Support Fund | Athina Georgiadou |
| Imperial College London | Imperial College Dean's EPSRC Studentship | Claire Dunican |
| UK Department for International Development, and European Union EDCTP2 program | MR/L006529/1 | Aubrey J Cunnington |
| Wellcome Trust | 206508/Z/17/Z | Myrsini Kaforou |

The funders had no role in study design, data collection and interpretation, or the decision to submit the work for publication.

### Author contributions

Athina Georgiadou, Conceptualization, Formal analysis, Investigation, Methodology, Writing – original draft, Writing – review and editing; Claire Dunican, Formal analysis, Writing – original draft, Writing – review and editing; Pablo Soro-Barrio, Hyun Jae Lee, Formal analysis, Writing – review and editing; Myrsini Kaforou, Writing – original draft, Writing – review and editing; Aubrey J Cunnington,

Conceptualization, Funding acquisition, Supervision, Writing – original draft, Writing – review and editing

**Author ORCIDs**
Athina Georgiadou [ID] http://orcid.org/0000-0002-0440-9335
Pablo Soro-Barrio [ID] http://orcid.org/0000-0003-4509-6109
Myrsini Kaforou [ID] http://orcid.org/0000-0001-9878-4007
Aubrey J Cunnington [ID] http://orcid.org/0000-0002-1305-3529

**Ethics**
All protocols and procedures were approved by Imperial College Animal Welfare and Ethical Review Board, following Laboratory Animal Science Association good practice guidance.

**Decision letter and Author response**
Decision letter https://doi.org/10.7554/eLife.70763.sa1
Author response https://doi.org/10.7554/eLife.70763.sa2

---

# Additional files

**Supplementary files**
• Supplementary file 1. Supplementary Figures and small tables. (A): The proportions of different leukocyte subtypes in the Lee et al. RNA-Seq data vs. the mouse models. (B): Details of the publicly available microarray data sets. (C): Comparison of host differential gene expression at onset of symptoms in a controlled human malaria infection study and at early stage illness in five mouse malaria models. (D): The Discordance-concordance analysis of the Boldt et al. Gabonese CH-UM vs Lee et al. Gambian children CM-UM differential expression analysis. (E): Gating strategy for defining WBC proportions in mouse blood. (F): Leucocyte proportions measured in whole blood by flow cytometry. (G): The genetic confirmation of the identity of the two different *Plasmodium yoelii* strains using the RNA-Seq reads in conjunction with the known single nucleotide variant. (H): The genetic confirmation of the two different *Plasmodium berghei* strains using the RNA-Seq reads in conjunction with four known single nucleotide variants.

• Supplementary file 2. Mouse Differential Expression Analysis without adjustment for immune cell type proportions.

• Supplementary file 3. *Boldt et al., 2019* Differential Expression Analysis without immune cell type adjustment.

• Supplementary file 4. *Idaghdour et al., 2012* Differential Expression Analysis without cell type adjustment.

• Supplementary file 5. The results of the CHMI microarray data set differential expression analysis without cell type mixture adjustment.

• Supplementary file 6. *Idaghdour et al., 2012* Differential Expression Analysis.

• Supplementary file 7. *Boldt et al., 2019* Differential Expression Analysis.

• Supplementary file 8. Human Mouse Orthologs.

• Supplementary file 9. Discordance Concordance.

• Supplementary file 10. The results of the CHMI microarray data set differential expression analysis.

• Supplementary file 11. GO terms.

• Supplementary file 12. Mouse Differential Expression Analysis.

• Supplementary file 13. Euclidean Distances.

• Supplementary file 14. *Lee et al., 2018b* Differential Expression Analysis.

• Supplementary file 15. *Lee et al., 2018b* Differential Expression Analysis without adjustment.

• Supplementary file 16. The results of the microarray mouse PcAS differential expression analysis.

• Supplementary file 17. Severity scoring protocols.

• Supplementary file 18. RNA-Seq Mouse Raw Counts.

• Supplementary file 19. Mouse Cell Type Proportions.

• Supplementary file 20. Metadata Sample IDs Phenotypes.

• Supplementary file 21. *Lee et al., 2018b* Sample Metadata.

• Supplementary file 22. Reference Immune Cell Type Expression Profile for Human Microarray data sets.

• Supplementary file 23. PCA input standardized logFC values.

• Supplementary file 24. Genes contributing to PC1 and PC2.

• Supplementary file 25. GO Term Background Gene Lists.

• Transparent reporting form

### Data availability

Mouse sequencing data available at ENA under the ID PRJEB43641. All data generated or analysed in this study are included in the manuscript and supplementary files 1-25.

The following dataset was generated:

| Author(s) | Year | Dataset title | Dataset URL | Database and Identifier |
|---|---|---|---|---|
| Georgiadou et al. | 2021 | Comparative transcriptomics reveals translationally relevant processes in mouse models of malaria | https://www.ebi.ac.uk/ena/browser/view/PRJEB43641 | ENA, PRJEB43641 |

The following previously published datasets were used:

| Author(s) | Year | Dataset title | Dataset URL | Database and Identifier |
|---|---|---|---|---|
| Boldt et al. | 2019 | Whole blood transcriptome of childhood malaria | https://www.ncbi.nlm.nih.gov/geo/query/acc.cgi?acc=GSE1124 | NCBI Gene Expression Omnibus, GSE1124 |
| Idaghdour et al. | 2012 | The genomic architecture of host whole blood transcriptional response to malaria infection | https://www.ncbi.nlm.nih.gov/geo/query/acc.cgi?acc=GSE34404 | NCBI Gene Expression Omnibus, GSE34404 |
| Lee et al. | 2018 | Dual RNA-seq of peripheral blood from Gambian children with severe or uncomplicated Plasmodium falciparum malaria | https://www.ebi.ac.uk/arrayexpress/experiments/E-MTAB-6413/ | ArrayExpress, E-MTAB-6413 |
| Milne et al. | 2021 | Longitudinal profiling of the human immune response to Plasmodium falciparum | https://www.ncbi.nlm.nih.gov/geo/query/acc.cgi?acc=GSE132050 | NCBI Gene Expression Omnibus, GSE132050 |
| Talavera-López et al. | 2019 | Whole blood transcriptome during acute phase of infecion in avirulent and virulent Plasmodium chabaudi chabaudi infection | https://www.ncbi.nlm.nih.gov/geo/query/acc.cgi?acc=GSE93631 | NCBI Gene Expression Omnibus, GSE93631 |

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
