## [Editor Report]

Using comparative transcriptomics, the authors performed an unbiased investigation of similarities and differences of mouse malaria models and human *Plasmodium falciparum* malaria. Whilst the data cannot convincingly identify which mouse models are best suited for studying specific human malaria phenotypes, the comparative analyses do indicate that these models reflect the broad diversity of human malaria disease. These comparative analyses provide a scientific rationale for the use of rodent malaria models.

---

## [Decision Letter]

**Decision letter after peer review:**

Thank you for submitting your article "Comparative transcriptomic analysis reveals translationally relevant processes in mouse models of malaria" for consideration by *eLife*. Your article has been reviewed by 3 peer reviewers, and the evaluation has been overseen by a Reviewing Editor and Bavesh Kana as the Senior Editor. The following individual involved in review of your submission has agreed to reveal their identity: Laurent Renia (Reviewer #1).

The reviewers have discussed their reviews with one another, and the Reviewing Editor has drafted a decision to help you prepare a revised submission. Please also note that reviewers felt that your article would be more suitable as a Resource paper at *eLife*. We suggest that you consider this when revising and resubmitting the manuscript.

Essential revisions:

This study represents a comparison of transcriptomic changes in malaria-infected mice versus humans to determine whether and which mouse malaria model(s) correspond to human malaria responses. Owing to the long-standing controversy regarding the relevance of mouse models to human disease pathogenesis and immune responses, the underlying purpose of the study is valuable.

1) The authors should more clearly inform readers of fundamental differences between the mouse models and human malaria infections they study that will contribute to differences, including:

– Humans but not mice are the natural host for their corresponding parasites.

– Human infection is initiated by sporozoite inoculation vs blood stage in their models.

– Human Pf densities are far below the values seen in the Py17XL mouse model.

– Humans here are malaria-experienced while mice are malaria-naïve in this study.

2) An essential aspect is that the models used need to be extremely well defined (mouse strain (sex and age,.), parasite origins and mode of propagation, etc..) and referenced.

3) The transcriptomics data should be more carefully analyzed and discussed for all the possible confounding factors. For example, reporting of the analyses should be expanded to better evaluate the inferences, further dataset comparisons are warranted and should not be difficult (Lee vs Boldt human datasets; early mice vs human CHMI; perhaps severe to non-severe mice).

Provide a better comparison population for UM. For example, please to add a public CHMI dataset in naïve humans. Pre-severe (i.e., early) is not the same as UM. Also, please add some public mouse data.

4) Much of the signal is an artifact of adding covariates for cell-populations to their models-especially since flow-based vs. inferred abundances are completely confounded with mouse/human. The effect of correcting for cell counts in the mice versus by transcriptomic deconvolution in human datasets should be made clearer in the methods, as the current wording on the human datasets is confusing by stating these "had already been adjusted for variation in leukocyte proportions".

5) The top gene lists and PCs (pathway enrichments) are not coherent. The heatmaps of the genes driving the PCA plots/pathways need to be shown as well as the extended list of pathway enrichments.

6) Outline the existing dogma about which mouse models best represent particular human syndromes, the hypothesized pathogeneses which are or are not seen in mouse models. The pathways/processes in their comparisons should be addressed. This might also help with the concerns about the Py17XL conclusions; while the Py17XL is interesting, these claims should be toned down.

7) Provide correct labels for supplemental material.

*Reviewer #1 (Recommendations for the authors):*

Additional points

1. Definition of the different mouse models: The authors should describe in more detail their parasites and mouse strain used. This is essential since it has been shown in many studies that different host/parasite combinations may lead to different infections or pathological outcomes. The authors should give more details on the origin of their parasites (provide references); Do consider a PCR or some other method to confirm that the different Plasmodium strains used are correct. This will contribute to the scientific rigor of the manuscript.

Are these cloned lines? Please state how they were propagated, i.e., through mosquitoes or blood transfer? Were they passaged in the same mouse strain? It has been shown that passaging the parasites through different mouse strains alters their virulence (Amani et al., Infect Immun, 1990).

Also, there are conflicting reports for P. yoelii 1XNL. Some labs have reported that this parasites strain is not uniformly lethal in B6 mice, contrary to BALB/c mice (Miyakoda et al., Front Immunol, 2018; Azcarate et al., Plos One, 2014). This may also depend on the dose of parasites and the site of injections used to initiate the infections. To add to the discussion, the best comparison should have been done with outbred mice such as Swiss mice, to mimic what happens in humans., and infection through mosquito bites.

2. The P yoelii 17XL is an interesting model for hyperlactatemia, a pathology that has been understudied. However, the cerebral microvascular pathology in PY17XL is not clearly defined as proposed. There is no evidence that Py17XNL cytoadhere to endothelial cells and sequester (an essential pathway in Human severe malaria and P. berghei ANKA).

3. Precise if the B6 mice use are B6/J or B6/N.

4. A missing analysis is with a comparison between the Gambian and the Gabonese. Are they concordant? And if not in which pathways.

*Reviewer #2 (Recommendations for the authors):*

Given the points raised above in the public review, the authors should justify (or adjust) their decision to compare malaria-naïve mice to malaria-experienced humans, rather than to malaria-naïve humans for whom numerous transcriptomes have been reported from controlled infection studies-to compare responses to "uncomplicated human malaria" vs "early mouse infection timepoints". This could contribute the poor concordance by PCA.

For comparisons of "uncomplicated malaria"/early timepoints, the PCA shows poor concordance for any model, whereas the most differentially expressed human genes show some correspondence to the different mouse models. "Severe malaria" shows the converse: PCA shows strong concordance for Py17XL with some human syndromes, but poor concordance for the 20 most differentially expressed human genes. The authors should better explain this apparent discrepancy in the results, which raises questions about the robustness of their inferences and approach.

Specific comments:

Title: "translationally relevant processes" is an awkward construction and will be confusing in a title that includes the word "transcriptomic".

L27 "between species and across all models" Please clarify for readers whether species refers to human vs mouse and not "between parasite species".

L 52 "Recently it has been proposed" Probably fairer to readers if the authors state "Recently we proposed" or "we and others".

L85 "five rodent malaria parasite species"-I think 3 species (Py, Pb, Pc) are studied here, with two lines/strains each representing Py and Pb.

L 142 "mouse models varied from 58 to 73% concordance" This sounds pretty good, but the authors should make it easier for readers to understand (in the main narrative) how they arrived at the subset of genes included in each analysis, which varied between comparisons (when one digs into File S6, which is labeled ""95442_0_supp_2049414_q3f8vp). Can the authors assign a p value in some way to tell us how likely this may have been due to chance? File S6 has spreadsheets counts/proportions of mouse transcripts aligned to 3 human studies, which indicate that these proportions considered only the genes that were either concordant ("same direction") and discordant ("different direction"). How were genes with minimal change in the mice counted, and what proportion of these genes that really showed no meaningful change in the mice were nevertheless assigned a count in one direction or the other?

Lines 165-170 "PC1 showed enrichment…" it's not clear how the authors distilled the long list of GO terms in File S7 to report the few responses they list as driving PC1 and PC2 in their PCA analysis. Authors should clarify this.

Figure 1-do the authors have data on anemia in their models that can be provided? This is relevant to their comparison of the models to cases of severe anemia. If no data, they could helpfully provide information from the literature where available.

*Reviewer #3 (Recommendations for the authors):*

– PCA plots of only the first 2 PCs were used to assess similarities and differences. GO enrichments were only performed on the genes driving these first 2 PCs. So, if PC1 in Figure 1B reflects a range of differences in 'enrichment of leukocyte immunity and adaptive immune response', from human on the left, PbNK65 and Py17XL in the middle, and PbANKA/Py17XNL/PcAS on the far right, what does this mean in terms of this GO for each cluster?

– It's helpful to see the top 20 DE genes in human in a heatmap with mouse, but the PCs are driven by different genes, and it's not clear if only these 10 were used for pathway enrichments, and how they compare across models and species. The largest drivers of variation may actually be less interesting than lower PCs b/c they may just be reflecting cell population corrections.

– To assess DE in severe disease, the authors compared early vs. late timepoints, citing PMID: 29695497. This review highlights that many changes are already present at early timepoints. This is concerning because DE in the human studies was done by comparing μm vs. SM, which are different disease classes (i.e., μm is not that same as pre-symptomatic).

– The overarching conclusion seems to be that Py17XL is the best match for all human studies. Much has been published on which mouse models are canonical for specific human syndromes, and where there are similarities and differences between our understanding of pathogenesis. It would add considerable values to this paper to include a summary of the key pathways in these mouse models, where they agree/disagree with human pathogenesis hypotheses, and then highlight these specific pathways in your analyses.

---

## [Author Response]

Essential revisions:This study represents a comparison of transcriptomic changes in malaria-infected mice versus humans to determine whether and which mouse malaria model(s) correspond to human malaria responses. Owing to the long-standing controversy regarding the relevance of mouse models to human disease pathogenesis and immune responses, the underlying purpose of the study is valuable.1) The authors should more clearly inform readers of fundamental differences between the mouse models and human malaria infections they study that will contribute to differences, including:– Humans but not mice are the natural host for their corresponding parasites.– Human infection is initiated by sporozoite inoculation vs blood stage in their models.– Human Pf densities are far below the values seen in the Py17XL mouse model.– Humans here are malaria-experienced while mice are malaria-naïve in this study.

We have added these points in the introduction (lines 84-90), have added data from a comparison between early stages of mouse infection and controlled human malaria infection in malaria naïve individuals (lines 188-192) and expanded discussion about differences in parasite load between the Py17XL infection in mice and severe malaria in humans (lines 496-502)

2) An essential aspect is that the models used need to be extremely well defined (mouse strain (sex and age,.), parasite origins and mode of propagation, etc..) and referenced.

We have clarified the strain (C57BL/6J), sex, and age of mice throughout the text and figure legends. We have expanded the methods section, explaining and referencing the origin and propagation of the malaria parasites we used in our experiments (lines 545-547) and explaining how we confirmed the authenticity of these parasite strains and that they had not become contaminated with other parasite strains (lines 653-672, Supplementary File 1G, 1H) by analysis of RNA-Seq read mapping to reference genomes and identifying the nucleotide mismatches consistent with literature-derived SNPs that distinguish between related parasite strains of *P. yoelii* and *P. berghei*.

3) The transcriptomics data should be more carefully analyzed and discussed for all the possible confounding factors. For example, reporting of the analyses should be expanded to better evaluate the inferences, further dataset comparisons are warranted and should not be difficult (Lee vs Boldt human datasets; early mice vs human CHMI; perhaps severe to non-severe mice).Provide a better comparison population for UM. For example, please to add a public CHMI dataset in naïve humans. Pre-severe (i.e., early) is not the same as UM. Also, please add some public mouse data.

We have added analysis of a dataset from CHMI in previously malaria-naïve individuals to the early-uninfected comparison (explained in lines 153-157; 188-192; 724-726 Supplementary File 1B, 1C). Interestingly the major inferences about relationships between early stages of infection in mouse models and in human malaria remain very similar to those derived from the uncomplicated malaria subjects in endemic populations.

We undertook an extensive search of the available literature for public datasets of the mouse blood transcriptome at comparable stages of infection. There is actually very little such data available for re-analysis. We identified microarray data from suitable stages of *P. chabaudi* AS infection in a small number of mice, and have included this in comparison to the *P. chabaudi* AS transcriptome data which we generated using RNA-Seq (lines 391-395; 731-737; Supplementary Files 2, 9, 16). Reassuringly there was high (86%) concordance between these independent mouse datasets.

We have added an assessment of the concordance of findings in subjects with similar SM phenotypes in the Lee et al., (RNA-Seq) dataset and Boldt et al., (microarray) datasets (lines 363-365; Supplementary File 1D, Supplementary File 9). Reassuringly there was high (78%) concordance between these phenotypes in datasets collected independently in different patient populations and generated using different platforms.

Throughout the manuscript we have highlighted issues which may confound comparisons and interpretation of the gene expression studies included in our analyses, and the approaches we have taken to minimize their impact (lines 130-141; 159-163; 331-334; 353-360; 471-478).

4) Much of the signal is an artifact of adding covariates for cell-populations to their models-especially since flow-based vs. inferred abundances are completely confounded with mouse/human. The effect of correcting for cell counts in the mice versus by transcriptomic deconvolution in human datasets should be made clearer in the methods, as the current wording on the human datasets is confusing by stating these "had already been adjusted for variation in leukocyte proportions".

We have provided additional justification of the adjustment for leukocyte proportions (line 159-163; 603-607; Supplementary File 1A). The changes in cell proportions induced by infection within species are large, and the differences in cell proportions in blood between mouse and human are even larger (Supplementary File 1A). Failure to account for this would results in cell mixture being the main determinant of all differences in gene expression within and between species.

We have modified the description of deconvolution and adjustment in the Lee et al., RNA Seq dataset to clarify that this was performed as described in the original publication.

We appreciate that there is not universal agreement in the transcriptomics field about the need to adjust for cell proportions, and so we have presented results of differential gene expression analyses without adjustment to strengthen the value of this resource (Supplementary Files 2, 3, 4, 5, 15).

5) The top gene lists and PCs (pathway enrichments) are not coherent. The heatmaps of the genes driving the PCA plots/pathways need to be shown as well as the extended list of pathway enrichments.

In the main figures, the principal component analyses and the heatmaps showing the comparisons of the “top” human genes with their orthologs in mice provide two different perspectives on the data and are not necessarily expected to be completely concordant (coherent) with one another. These different approaches to evaluating how well the mouse models recapitulate the changes in gene expression in humans are introduced and explained in the Results section (lines 167-179).

For each PCA plot in the main figures, we have added figure supplements (for figures 2, 3, 4) showing full heatmaps for the expression of genes contributing most to the first two principal components in humans and each mouse model. Further explanation has been added to the methods section (lines 806-811)

6) Outline the existing dogma about which mouse models best represent particular human syndromes, the hypothesized pathogeneses which are or are not seen in mouse models. The pathways/processes in their comparisons should be addressed. This might also help with the concerns about the Py17XL conclusions; while the Py17XL is interesting, these claimsshould be toned down.

We have added some additional explanation of the current understanding of the models used in this study and their relationship to human disease in the introduction (Lines 69-78). We have added comments in the results about the relevance of the top pathways (GO terms) identified in our analyses and what is known in human and mice (lines 201-208, 211-213, 296-300, 381-384) We have added further discussion to ensure that our conclusions about the relevance of Py17XL are not overinterpreted (lines 496-498) and to highlight the issue of differences in parasite load in human SM and Py17XL infection (lines 498-502).

7) Provide correct labels for supplemental material.

All supplementary materials have been correctly labelled in their file names and within the text of each file.

Reviewer #1 (Recommendations for the authors):Additional points1. Definition of the different mouse models: The authors should describe in more detail their parasites and mouse strain used. This is essential since it has been shown in many studies that different host/parasite combinations may lead to different infections or pathological outcomes. The authors should give more details on the origin of their parasites (provide references); Do consider a PCR or some other method to confirm that the different Plasmodium strains used are correct. This will contribute to the scientific rigor of the manuscript.

This has been addressed above in response to the essential revisions

Are these cloned lines? Please state how they were propagated, i.e., through mosquitoes or blood transfer? Were they passaged in the same mouse strain? It has been shown that passaging the parasites through different mouse strains alters their virulence (Amani et al., Infect Immun, 1990).

This has been addressed above in response to the essential revisions

Also, there are conflicting reports for P. yoelii 1XNL. Some labs have reported that this parasites strain is not uniformly lethal in B6 mice, contrary to BALB/c mice (Miyakoda et al., Front Immunol, 2018; Azcarate et al., Plos One, 2014). This may also depend on the dose of parasites and the site of injections used to initiate the infections. To add to the discussion, the best comparison should have been done with outbred mice such as Swiss mice, to mimic what happens in humans., and infection through mosquito bites.

We have added this to the discussion

2. The P yoelii 17XL is an interesting model for hyperlactatemia, a pathology that has been understudied. However, the cerebral microvascular pathology in PY17XL is not clearly defined as proposed. There is no evidence that Py17XNL cytoadhere to endothelial cells and sequester (an essential pathway in Human severe malaria and P. berghei ANKA).

This is addressed in part in the discussion about parasite load and sequestration in response to the essential revisions

3. Precise if the B6 mice use are B6/J or B6/N.

This has been clarified

4. A missing analysis is with a comparison between the Gambian and the Gabonese. Are they concordant? And if not in which pathways.

This has been addressed above in response to the essential revisions

Reviewer #2 (Recommendations for the authors):Given the points raised above in the public review, the authors should justify (or adjust) their decision to compare malaria-naïve mice to malaria-experienced humans, rather than to malaria-naïve humans for whom numerous transcriptomes have been reported from controlled infection studies-to compare responses to "uncomplicated human malaria" vs "early mouse infection timepoints". This could contribute the poor concordance by PCA.

This has been addressed above in response to the essential revisions

For comparisons of "uncomplicated malaria"/early timepoints, the PCA shows poor concordance for any model, whereas the most differentially expressed human genes show some correspondence to the different mouse models. "Severe malaria" shows the converse: PCA shows strong concordance for Py17XL with some human syndromes, but poor concordance for the 20 most differentially expressed human genes. The authors should better explain this apparent discrepancy in the results, which raises questions about the robustness of their inferences and approach.

This is not due to a discrepancy in the results or a lack of reproducibility. It is simply not possible to make these inferences directly from the PCA plots because the distances between human and mouse infections in different plots cannot be compared. The relative distances within each plot indicate the similarity of the changes in gene expression of the samples included in that plot only. Relative distances cannot be compared between plots because they are effectively on different scales.

We believe this is a consequence of the transmission of files from bioRxiv and we have updated all file names as addressed above in response to the essential revisions

Specific comments:Title: "translationally relevant processes" is an awkward construction and will be confusing in a title that includes the word "transcriptomic".

We believe the title conveys the purpose of this project and would prefer to keep it as it is.

L27 "between species and across all models" Please clarify for readers whether species refers to human vs mouse and not "between parasite species".

This has been clarified “between the different host species and across all models”

L 52 "Recently it has been proposed" Probably fairer to readers if the authors state "Recently we proposed" or "we and others".

This has been changed to: "we and others"

L85 "five rodent malaria parasite species"-I think 3 species (Py, Pb, Pc) are studied here, with two lines/strains each representing Py and Pb.

This has been changed to: “The five rodent malaria parasite strains”

L 142 "mouse models varied from 58 to 73% concordance" This sounds pretty good, but the authors should make it easier for readers to understand (in the main narrative) how they arrived at the subset of genes included in each analysis, which varied between comparisons (when one digs into File S6, which is labeled ""95442_0_supp_2049414_q3f8vp). Can the authors assign a p value in some way to tell us how likely this may have been due to chance? File S6 has spreadsheets counts/proportions of mouse transcripts aligned to 3 human studies, which indicate that these proportions considered only the genes that were either concordant ("same direction") and discordant ("different direction"). How were genes with minimal change in the mice counted, and what proportion of these genes that really showed no meaningful change in the mice were nevertheless assigned a count in one direction or the other?

Since the concordance-discordance analyses were intended to primarily be descriptive we have not calculated p-values, but one would expect 50% concordance to occur by chance. We have included an explanation of how these were calculated in the methods (lines 779-787). We chose not to stipulate a log-fold change in the mice in the main analysis, because we did not have any clear justification for a particular threshold and the relatively small number of mice in each group would potentially result in many genes with small but important changes in gene expression being excluded.

Lines 165-170 "PC1 showed enrichment…" it's not clear how the authors distilled the long list of GO terms in File S7 to report the few responses they list as driving PC1 and PC2 in their PCA analysis. Authors should clarify this.

Since GO terms have hierarchical relationships it is always challenging to determine which terms to report. We tried to select terms from amongst those with the smallest enrichment p-values which conveyed some specificity of the biological process (eg. “response to bacteria” rather than “response to organism”). Presenting these GO terms as a resource will allow others to look at the level of detail which is of greatest interest to them.

Figure 1-do the authors have data on anemia in their models that can be provided? This is relevant to their comparison of the models to cases of severe anemia. If no data, they could helpfully provide information from the literature where available.

Data on anaemia in Py17XL is presented in Figure 5. We have added references to the anaemia occurring in each of the infections in the introduction (Lines 74-78).

Reviewer #3 (Recommendations for the authors):– PCA plots of only the first 2 PCs were used to assess similarities and differences. GO enrichments were only performed on the genes driving these first 2 PCs. So, if PC1 in Figure 1B reflects a range of differences in 'enrichment of leukocyte immunity and adaptive immune response', from human on the left, PbNK65 and Py17XL in the middle, and PbANKA/Py17XNL/PcAS on the far right, what does this mean in terms of this GO for each cluster?

The gene ontology terms provide a “flavour” of what is driving the separation on the PC plots, but we believe that the heatmaps of the individual genes driving each PC are more instructive in this respect. These have been added as supplementary figures for each main figure, as addressed above in response to the essential revisions. We have not described each of these in detail in the main text because we fear this would confuse many readers interested in the big picture, but it is clear that for some genes in the example above, such as *VNN1*, there is a gradient of expression consistent with the separation across the PCA plot, whilst for others there is a more heterogeneous pattern.

– It's helpful to see the top 20 DE genes in human in a heatmap with mouse, but the PCs are driven by different genes, and it's not clear if only these 10 were used for pathway enrichments, and how they compare across models and species. The largest drivers of variation may actually be less interesting than lower PCs b/c they may just be reflecting cell population corrections.

In the methods section we explain that all genes contributing greater than or equal to 0.1% to PC1 and/or PC2 were used in gene ontology analysis. Due to the limited space, we only show the top 10 genes in the mini-heat maps parallel to the axes in the PCA plots, to give readers an idea of the main drivers.

– To assess DE in severe disease, the authors compared early vs. late timepoints, citing PMID: 29695497. This review highlights that many changes are already present at early timepoints. This is concerning because DE in the human studies was done by comparing μm vs. SM, which are different disease classes (i.e., μm is not that same as pre-symptomatic).

In our mouse experiments we collected blood from mice at onset of physical signs of illness, rather than prior to onset of illness, but we accept that this stage of infection may not be equivalent to human UM. For this reason we have now included a human CHMI dataset in which we have data from humans at first onset of symptoms or signs of infection. This has been addressed in more detail above in response to the essential revisions

– The overarching conclusion seems to be that Py17XL is the best match for all human studies. Much has been published on which mouse models are canonical for specific human syndromes, and where there are similarities and differences between our understanding of pathogenesis. It would add considerable values to this paper to include a summary of the key pathways in these mouse models, where they agree/disagree with human pathogenesis hypotheses, and then highlight these specific pathways in your analyses.

This has been addressed above in response to the essential revisions